# Teaching LLMs to Plan: Logical Chain-of-Thought Instruction Tuning for Symbolic Planning

## Abstract

Large language models (LLMs) have demonstrated impressive capabilities across diverse tasks, yet their ability to perform structured symbolic planning remains limited, particularly in domains requiring formal representations like the Planning Domain Definition Language (PDDL). In this paper, we present a novel instruction tuning framework, PDDL-Instruct, designed to enhance LLMs' symbolic planning capabilities through logical chain-of-thought reasoning. Our approach focuses on teaching models to rigorously reason about action applicability, state transitions, and plan validity using explicit logical inference steps. By developing instruction prompts that guide models through the precise logical reasoning required to determine when actions can be applied in a given state, we enable LLMs to self-correct their planning processes through structured reflection. Unlike prompting-based neuro-symbolic approaches, PDDL-Instruct updates model parameters directly using verification feedback, enabling more efficient learning from external validators. Experimental results on multiple planning domains show that our chain-of-thought reasoning based instruction-tuned models are significantly better at planning, achieving planning accuracy of up to 94% on standard benchmarks, representing a 66% absolute improvement over untuned baseline models. This work bridges the gap between the general reasoning capabilities of LLMs and the logical precision required for automated planning, offering a promising direction for developing better AI planning systems.

## 1 Introduction

Large Language Models (LLMs) like GPT (OpenAI et al., 2023), Gemini (Gemini Team et al., 2023), LLaMA (Touvron et al., 2023), etc. have demonstrated remarkable success across various domains including mathematics and coding (Imani et al., 2023; Gaur & Saunshi, 2023; Romera-Paredes et al., 2023; Ahn et al., 2024). However, a critical gap remains in their ability to perform structured symbolic planning – a fundamental capability required for reliable real-world sequential decision-making systems. Recent studies have highlighted this issue that while LLMs excel at general reasoning over unstructured text, they struggle with the logical reasoning and systematic verification required for automated planning tasks (Stechly et al., 2023; Valmeekam et al., 2023a;c; Kambhampati et al., 2024; Stechly et al., 2025).

This limitation becomes particularly evident when considering formal planning representations such as the Planning Domain Definition Language (PDDL) (McDermott et al., 1998). Despite some promising results with specific configurations (Liu et al., 2023; Wang et al., 2024), these models generally perform poorly on multi-step reasoning tasks including classical planning (Hsiao et al., 2025). This has significant implications for planning tasks, which are PSPACE-complete (Bylander, 1991) and inherently require scaling reasoning efforts with problem complexity.

Recent work has explored neuro-symbolic approaches that combine LLMs with external verifiers in a Generate-Test-Critique loop (LLM-Modulo framework (Kambhampati et al., 2024)). While these methods improve plan validity through prompting-based feedback, they suffer from a critical limitation that the feedback is used only at inference time to guide generation, not to update model parameters. This means LLMs must re-learn correct planning principles through multiple feedback

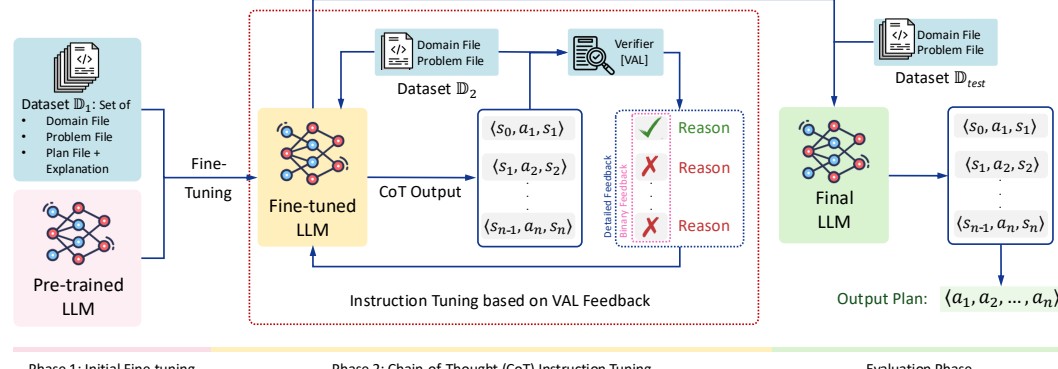

Figure 1: The PDDL-INSTRUCT approach consists of three phases: Two training phases (Initial and CoT Instruction Tuning) and evaluation phase. The main innovation lies in the second phase: CoT Instruction Tuning (highlighted by the red boundary). The initially tuned LLM is further trained using a structured instruction process that emphasizes complete logical reasoning chains.

iterations, making the approach computationally expensive and potentially unstable to self-correction failures (Stechly et al., 2025). We propose a different direction of using verification feedback to directly update model parameters during training.

In this paper, we introduce PDDL-INSTRUCT, a novel framework shown in Fig. 1, that augments LLMs' reasoning capabilities with the formal reasoning required for automated planning. PDDL-INSTRUCT explicitly teaches LLMs to reason through the precondition-effect structure of planning domains using logical chain-of-thought prompting. Rather than attempting to compete with symbolic planners which achieve 100% accuracy on these problems, our goal is to improve how LLMs learn to reason about planning through systematic instruction tuning and verification feedback. This foundational improvement in LLM planning capabilities is a necessary step toward developing hybrid systems that leverage both LLM flexibility and symbolic verification.

Main contributions of this paper are:

- A parameter-update-based instruction tuning framework that enhances symbolic planning in LLMs through logical chain-of-thought reasoning.

- A formalization of planning verification as decomposable reasoning chains with domain-informed loss functions, enabling LLMs to systematically check preconditions, apply effects, and validate invariants. This allows gradient-based optimization to directly target planning semantics rather than generic language modeling objectives.

- Empirical evidence that parameter-update-based instruction-tuned LLMs develop robust, generalizable planning capabilities within a single multi-domain model, without domain-specific retraining, achieving 94% accuracy on BlocksWorld and generalizing to unseen domains with 8-9% relative degradation.

Our results show that PDDL-INSTRUCT significantly outperforms both baseline models and traditionally instruction-tuned models, achieving planning validity rates of up to 94% in standard planning domains. This work not only addresses a critical limitation in current LLM capabilities but also provides a foundation for developing more trustworthy AI systems capable of reliable planning. in complex scenarios.

## 2 RELATED WORK

**LLMs for Planning**  The use of LLMs for planning has received a lot of attention (Pallagani et al., 2024). Various approaches have been used so far, such as dictating the planned behaviors by generating executable code (Liang et al., 2023; Singh et al., 2023; Nijkamp et al., 2023; Wang et al., 2025) or behavior trees (Zhou et al., 2024a; Izzo et al., 2024; Ao et al., 2025), using closed loop with environment feedback (Huang et al., 2022; Song et al., 2023; Sun et al., 2023) or for self-refinement

(Wang et al., 2023; Zhou et al., 2024b). A few recent approaches also synthesize Python programs using LLMs for planning (Silver et al., 2024; Hao et al., 2025b; Chen et al., 2025b; Hu et al., 2025; Chi et al., 2025). A complementary research direction explores using LLMs for parts of the search process, like generating heuristics (Ahn et al., 2022; Liu et al., 2024; Corrêa et al., 2025), reducing large search spaces (Zhao et al., 2023), predicting transition functions (Shlomi et al., 2025), etc.

However, as summarized in Tantakoun et al. (2025), LLMs face challenges with long-term planning and reasoning, often producing unreliable plans (Stechly et al., 2024; Pallagani et al., 2023; Momen-nejad et al., 2023), frequently failing to account for the effects and requirements of actions as they scale (Stechly et al., 2024), and their performance degrades with self-iterative feedback (Stechly et al., 2023; Valmeekam et al., 2023a; Huang et al., 2025b).

Finetuning for planning improves significantly the model's capabilities to generate symbolic plans (Pallagani et al., 2023; Li et al., 2025; Fu et al., 2025). However, the main drawbacks of this approach are its high economic, time, and computational costs, as well as the degradation of the transferability of the model. Finetuning makes the model specialized on the domains and problem types trained on, with poor transferability to new problems. An extended literature review on LLMs and Planning is available in the appendix.

**LLM Modulo Approaches**   A popular neuro-symbolic approach for using LLMs for symbolic tasks is to integrate them with symbolic verifier similar to a LLM Modulo framework (Kambhampati et al., 2024). Various approaches based on this framework, like CoT-TL (Manas et al., 2024), Code-as-Symbolic-Planner (Chen et al., 2025c), Planning using Neuro Symbolic Reasoning (Jha et al., 2023), LEPA (Zhang et al., 2025a), STaR (Zelikman et al., 2022), etc. use a verifier to give feedback on the wrong output generated by an LLM. While these approaches improve plan quality, they operate through prompting-based feedback loops at inference time. In contrast, PDDL-INSTRUCT uses verification feedback to update model parameters during training, enabling the model to learn robust planning reasoning rather than relying on repeated prompting. Additionally, our domain-informed loss functions (encoding precondition checking, effect application, goal verification) directly optimize planning semantics, whereas existing methods treat feedback as text for in-context learning.

**Instruction Tuning**   Instruction tuning has emerged as a significant approach in NLP to enable zero-shot generalization on unseen tasks (Mishra et al., 2022; Wei et al., 2022a; Ouyang et al., 2022). This technique involves fine-tuning large language models to perform diverse tasks by following instructions, making the task source crucial for effective tuning (Longpre et al., 2023). While existing methods often rely on human-crowdsourced tasks from datasets like T0 (Sanh et al., 2022), FLAN (Wei et al., 2022a; Longpre et al., 2023), and NaturalInstructions (Mishra et al., 2022; Wang et al., 2022), these high-quality resources demand significant human effort and are typically limited in quantity. An alternative approach involves model-generated tasks, where powerful language models like GPT-3 and GPT-4 generate diverse instructions and task pairs (Wang et al., 2022; Peng et al., 2023), though these can introduce noise when outputs don't properly correspond to inputs. In this work, we alleviate this problem by leveraging the automated planning task generators (Seipp et al., 2022; Valmeekam et al., 2023b) to create the instruction tuning dataset.

**Chain-of-Thought Reasoning**   A significant advancement in improving LLM reasoning ability is the implementation of Chain of Thought (CoT) prompting (Wei et al., 2022b). By generating explicit intermediate reasoning steps, these models can now address complex logical deduction and multistep problem-solving. Short CoT approaches (Lambert et al., 2025; Kojima et al., 2022) demonstrated effectiveness for straightforward problems but revealed limitations when confronting more intricate challenges. The evolution toward longer reasoning chains has subsequently transformed the landscape of machine reasoning. Stechly et al. (2024) argued that despite its efficacy for reasoning tasks, CoT is not suitable for planning, but in this work we show that with proper integration of instruction tuning using better prompts, CoT can indeed be used for planning tasks.

## 3    PRELIMINARIES

**Automated Planning**   In this section, we briefly describe automated planning. Please refer to Geffner & Bonet (2013) and Chen et al. (2025a) for more details.

An automated planning problem can be formally characterized as a tuple $\langle P, A, s_0, G \rangle$, where $P$ is a set of fluents used to describe a discrete and fully-observable state $S$, $A$ represents a finite set of actions, $s_0 \in S$ denotes the initial state, and $G$ specifies the goal conditions. Each action $a_i \in A$ is defined as $\langle pre(a_i), add(a_i), del(a_i) \rangle$, where $pre(a_i)$ is the set of fluents that must hold in the current state for the action to be executable, $add(a_i)$ is the set of fluents that become true after executing $a_i$, and $del(a_i)$ is the set of fluents that become false after executing $a_i$. Note that the state space $S$ in classical planning emerges from all possible truth assignments to the set of fluents.

A solution to a planning problem $\mathcal{P}$, called a plan $\pi$, is a sequence of actions $\langle a_1, a_2, ..., a_n \rangle$ that transforms the initial state into one satisfying the goal conditions after $n$ steps. Note that $\pi$ produces state transitions $s_{i+1} = a_{i+1}(s_i) = (s_i \setminus del(a_{i+1})) \cup add(a_{i+1})$ for all $0 \leq i < n$ such that $s_n \in G$. $\pi$ is considered *optimal* if it takes the least number of actions (in this work, we consider actions with uniform cost) to reach a goal state, whereas it is considered *satisficing* if it reaches the goal successfully but with more actions than needed by an optimal plan.

The Planning Domain Definition Language (PDDL) (McDermott et al., 1998), based on STRIPS (Fikes & Nilsson, 1971), provides a standardized specification for automated planning problems. PDDL consists of a *domain* $\mathcal{D} = \langle P, A \rangle$ containing the sets of fluents $P$ and actions $A$ (along with their precondition, $add$ and $del$ sets), and a *problem* $\mathcal{P} = \langle s_0, G \rangle$ containing the initial state $s_0$, and a goal condition $G$.

**Instruction Tuning** Instruction tuning (Mishra et al., 2022; Wei et al., 2022a; Ouyang et al., 2022) is an approach for fine-tuning LLMs on a labeled dataset. Consider an instruction tuning dataset $\mathbb{D}_1 = \{(x_i, \tau_i)\}_{i=1}^{\Omega}$ with $\Omega$ labeled samples, where $x_i$ represents an instruction and $\tau_i$ its corresponding ideal target response. We denote our large language model as $\mathcal{M}_\theta$ with parameters $\theta$. The model produces output $\mathcal{M}_\theta(x_i)$ for a given instruction $x_i$. The standard instruction tuning objective aims to find model parameters $\theta^*$ that minimize expected discrepancy (loss $\mathcal{L}$) between model predictions ($\mathcal{M}_\theta(x)$) and target responses ($\tau$) across the instruction dataset (Dataset $\mathbb{D}_1$, as described in Sec. 4):

$$\theta^* = \arg\min_\theta \mathbb{E}_{(x,\tau)\sim\mathbb{D}_1}[\mathcal{L}(\mathcal{M}_\theta(x), \tau)] \qquad (1)$$

**Chain-of-thought reasoning** Chain-of-Thought (CoT) reasoning can be formally defined as a structured decomposition of a complex reasoning task into an explicit sequence of intermediate logical steps. Given a problem input $x$ and a target output $y$, a chain-of-thought reasoning process $\mathcal{R}$ is a sequence of $K$ intermediate reasoning states $\mathcal{Z}(x) = (z_1, z_2, \ldots, z_K)$, where each $z_i$ represents an atomic reasoning step that transforms the latent state from $z_{i-1}$ to $z_i$, with $z_0$ implicitly defined as the initial problem state derived from $x$. Each reasoning step $z_i$ can be characterized as a tuple $z_i = (s_i, j_i, u_i)$, where $s_i$ represents the symbolic state (the set of derived facts or assertions at step $i$), $j_i$ represents the justification (the logical rule or inference applied), and $u_i$ represents the uncertainty estimate (the model's confidence in this reasoning step). For simplicity, going forward we will use symbolic states $s_i$ to represent reasoning states $z_i$, when clear from context, as they have a one-to-one mapping for this work. We also do not use $u_i$ estimates for this work, and the LLM is directly asked for the resulting symbolic states in each CoT step.

Two important properties that characterize effective chain-of-thought reasoning are: (i) logical coherence (Wei et al., 2022b), and (ii) progressive refinement (Du et al., 2025). A CoT process $\mathcal{R}(x)$ exhibits *logical coherence* if for each step $z_i$ with $i > 1$, $\exists j_{i-1}$ such that $j_{i-1}(s_{i-1}) \Rightarrow s_i$, meaning each state follows logically from the application of a justifiable inference rule to the previous state. A CoT process $\mathcal{R}(x)$ exhibits *progressive refinement* if $I(z_i; y) > I(z_{i-1}; y) \quad \forall i \in \{1, 2, ..., K\}$, where $I(z_i; y)$ represents the mutual information between reasoning state $z_i$ and the target output $y$.

## 4 PROBLEM FORMULATION

**Input** In this work, we use the following inputs: (i) a pre-trained LLM $\mathcal{M}$ as input, (ii) a dataset $\mathbb{D}$ of planning domains and problems expressed in PDDL with their solutions (satisficing plans), and (iii) a plan validator $\mathcal{V}$ used to validate the correctness of plans generated by $\mathcal{M}$. The dataset $\mathbb{D}$ consists of:

1. A set $\{\mathcal{D}_1, \mathcal{D}_2, ..., \mathcal{D}_n\}$ of planning domains expressed in PDDL.

2. For each domain $\mathcal{D}_i$, we have problems $\mathbb{P}_i = \{\mathcal{P}_{i,1}, \mathcal{P}_{i,2}, ..., \mathcal{P}_{i,m_i}\}$.

3. For each planning problem $P_{i,j}$, we have a valid plan $\Pi_{i,j} = \{\pi_{i,j,1}, \pi_{i,j,2}, ..., \pi_{i,j,k_{i,j}}\}$, where each plan $\pi_{i,j,l}$ is a sequence of grounded actions.

**Output**  The primary output is an instruction-tuned model $\mathcal{M}_{\theta*}$ with enhanced symbolic planning capabilities. The model should demonstrate improved domain representation, problem representation, plan generation, action verification, plan verification, and reasoning transparency.

**Assumptions**  Our framework assumes the planning domains follow the features explained in Sec. 3, i.e., does not contain complex PDDL features such as, e.g., conditional effects or durative actions. This simplifies the reasoning chain.

## 5 PDDL-INSTRUCT: METHODOLOGY

Fig. 1 illustrates our comprehensive framework for enhancing symbolic planning capabilities in Large Language Models (LLMs) through logical Chain-of-Thought (CoT) instruction tuning. The approach consists of two training phases: Initial Instruction Tuning and CoT Instruction Tuning.

### 5.1 TRAINING THE MODEL

**[Phase 1] Initial Instruction Tuning Phase**  In the initial instruction tuning phase (distinct from simple finetuning), we take a pre-trained LLM and train it with carefully crafted prompts that pair planning domains and problems with detailed explanations of their solutions, all derived from Dataset $\mathbb{D}_1$. As shown in Fig. 1, rather than simply exposing the model to planning examples, we explicitly instruct it to analyze why each action in a plan is valid by explaining precondition satisfaction and effect application.

This phase incorporates both correct plans and deliberately incorrect plans to teach the model to recognize and explain various planning errors. For incorrect plans, we include examples where: (1) action preconditions are not satisfied, (2) effects are incorrectly applied, (3) frame axioms are violated, or (4) the plan fails to reach the goal state. By exposing the model to both successful and failed planning attempts with detailed explanations, we establish a foundation for logical verification. The incorrect plans are generated by randomly replacing one of the actions in the correct plan with another action compatible with the problem. We verify using a plan validator, VAL (Howey et al., 2004), that this new plan is incorrect. We also add VAL's output for each plan (both correct and incorrect) to the dataset. For a subset of plans, we augmented the VAL explanations to create synthetic negative examples demonstrating incorrect effect application. Specifically, we generated failure scenarios where effects are incorrectly added or removed. This data augmentation strategy ensures the model learns to detect this particular class of planning errors, which VAL's output does not naturally produce.

This phase establishes a foundation of planning knowledge while simultaneously teaching the model to articulate logical justifications for action validity, setting the stage for more advanced reasoning in subsequent phases. Exact prompts used in this work are available in the supplementary material.

**[Phase 2] CoT Instruction Tuning Phase**  The main innovation of our approach lies in the CoT Instruction Tuning phase (highlighted by the red boundary in Fig. 1). This second phase is itself a two-stage process described thoroughly in the next section. At a high level, in this phase, the initially tuned LLM is further trained using a structured instruction process that emphasizes complete logical reasoning chains. When presented with a domain and problem from Dataset $\mathbb{D}_2$, this initially tuned model produces step-by-step state-action-state sequences $\langle s_0, a_1, s_1 \rangle, \langle s_1, a_2, s_2 \rangle, \ldots, \langle s_{n-1}, a_n, s_n \rangle$ that represent a candidate plan.

These reasoning chains are then passed through a verification module implemented using VAL (Howey et al., 2004) that systematically checks the validity of each state transition based on action preconditions and effects. Please note that while some approaches have tried using LLMs themselves as verifiers, research shows that currently LLMs do not possess sufficient self-correction capabilities in terms of reasoning (Huang et al., 2024; Stechly et al., 2025). Unlike self-reflection approaches where models attempt to critique their own reasoning without external validation, our chain-of-thought

method explicitly decomposes the planning process into verifiable logical steps, with external verification providing ground-truth feedback. This combination of explicit reasoning decomposition with verified feedback creates a more reliable foundation for enhancing planning capabilities than relying solely on the model's internal reasoning.

We explore two distinct types of verification feedback: (1) *binary feedback*, which simply indicates whether an action is valid or invalid, and (2) *detailed feedback*, which provides specific reasoning about each action generated by VAL in terms of which preconditions failed or which effects were incorrectly applied. Our hypothesis is that detailed feedback will lead to more robust planning capabilities by providing explicit guidance on the logical errors in the reasoning process.

The verification results provide crucial feedback that guides further instruction tuning. This feedback loop ensures that the model learns not only to generate syntactically correct plans but also to reason about their logical validity. We limit the number of times this feedback loop is used to generate new CoT plans, denoted by $\eta$. $\eta$ is a hyperparameter which we can vary to see how it affects accuracy.

Our PDDL-INSTRUCT approach prioritizes *logical coherence* (see Sec. 3) through its explicit verification of preconditions and effects at each planning step. The verification feedback ensures that each state transition follows logically from the application of a valid action, maintaining strict adherence to the domain rules. However, our approach does not ensure *progressive refinement* (see Sec. 3). This is because rather than optimizing for the shortest or most efficient plan (which would increase mutual information with an optimal solution at each step), we focus on producing satisficing plans that achieve the goal regardless of path length. Generating optimal solutions is a significantly more difficult problem in practice, both for classical planners and for training LLMs to produce them (Ray & Ginsberg, 2008; Domshlak & Nazarenko, 2013).

## 5.2 TRAINING METHODOLOGY FOR PHASE 2 CoT INSTRUCTION TUNING: OPTIMIZATION PROCESS

A distinctive feature of our PDDL-INSTRUCT framework is the two-stage optimization process as part of the CoT Instruction Tuning that explicitly targets both the quality of logical reasoning for CoT and the resulting final planning performance. This approach addresses the unique challenges of symbolic planning by ensuring that the model not only produces correct plans but also develops robust verification capabilities through logical chain-of-thought reasoning. An algorithm for this is available in the supplementary material.

**Stage 1: Reasoning Chain Optimization** In the first stage, we optimize the model parameters $\theta_t$ to improve the generation of high-quality reasoning chains. Specifically, the model weight in each reasoning step $r$, $\theta_t^r$ where $t \in [0, \eta - 1]$, is updated as Equation 2:

$$\theta_t^r = \theta_t - \delta_1 \nabla_{\theta_t} \mathcal{L}_{\text{reasoning}}(\theta_t, \mathbb{D}_{\text{reasoning}}^t) \tag{2}$$

where $\mathcal{L}_{\text{reasoning}}$ is a loss function that measures the quality of the generated reasoning chains compared to ideal logical inference sequences, $\delta_1$ is the learning rate for this stage, and $\mathbb{D}_{\text{reasoning}}^t$ is the dataset of individual $\langle s_{i-1}, a_i, s_i \rangle$ triplets along with VAL feedback for them. This objective encourages the model to produce step-by-step reasoning that correctly (i) checks all necessary preconditions before applying actions; (ii) tracks state changes resulting from action effects; (iii) verifies that invariants are maintained throughout the plan; and (iv) detects logical inconsistencies in proposed plans.

The reasoning loss explicitly penalizes logical errors such as applying actions with unsatisfied preconditions, failing to properly propagate effects, or generating steps that violate domain constraints. By focusing specifically on the reasoning process, this stage helps the model develop the logical foundation necessary for robust planning.

**Stage 2: End-Task Performance Optimization** In the second stage, we optimize from the reasoning-improved parameters $\theta_t^r$ to enhance overall planning:

$$\theta_{t+1} = \theta_t^r - \delta_2 \nabla_{\theta_t^r} \mathcal{L}_{\text{final}}(\theta_t^r, \mathbb{D}_{\text{final}}^t) \tag{3}$$

where $\mathcal{L}_{\text{final}}$ measures how well the final outputs match the expected answers in the training data, $\delta_2$ is the learning rate for this stage, and $\mathbb{D}_{\text{final}}^t$ final contains the domain, problem, and plan extracted from CoT output along with VAL feedback specifying if the plan is correct for that problem or not. This

second stage ensures that improvements in logical reasoning translate to practical planning capability of producing accurate plans.

This two-stage approach is important as Stage 1 develops the logical foundation needed for planning, while Stage 2 ensures these capabilities are properly applied to generate correct plans. The separation of these objectives allows our framework to balance between teaching fundamental reasoning skills and optimizing for task-specific performance, resulting in models that not only produce correct plans but can also reason about their correctness through explicit logical CoT inference. The exact formulations of the loss functions $\mathcal{L}_{\text{reasoning}}$ and $\mathcal{L}_{\text{final}}$ and the specific values of the hyperparameters are discussed in detail in the supplementary material.

### 5.3 EVALUATION PHASE

After completing both the Initial Instruction Tuning and CoT Instruction Tuning phases, the final model is evaluated in the Evaluation Phase (represented on the right side of Fig. 1). In this phase, the instruction-tuned LLM is presented with new, unseen planning domains and problems from $\mathbb{D}_{test}$. The model directly generates complete state-action-state sequences $\langle s_0, a_1, s_1 \rangle, \ldots, \langle s_{n-1}, a_n, s_n \rangle$ that constitute its proposed solution to the planning problem. These generated plans are then evaluated for correctness using VAL, but only for assessment purposes, i.e., no feedback is returned to the model. The plan is considered valid if and only if all actions in the sequence are applicable in their respective states and the final state satisfies all goal conditions.

## 6 EMPIRICAL EVALUATION

We conduct a comprehensive empirical evaluation of PDDL-INSTRUCT to assess its effectiveness in enhancing symbolic planning capabilities in LLMs. Our evaluation leverages PlanBench (Valmeekam et al., 2023b), a standardized benchmark framework for evaluating LLM planning capabilities.

We evaluate PDDL-INSTRUCT using PlanBench to assess its effectiveness in enhancing symbolic planning capabilities in LLMs. Our experiments aim to answer the following research questions:

**RQ1:** Does logical CoT instruction tuning improve plan validity compared to standard approaches?

**RQ2:** How does the quality of feedback (binary vs. detailed) affect planning performance?

**RQ3:** How well does the approach generalize across different planning domains?

**RQ4:** How robust are the models to problem complexity? Does performance degrade as it increases?

We implement PDDL-INSTRUCT using Llama-3-8B, GPT-4[1], and Gemma-3-270M (Gemma Team et al., 2025) models. We compare against baseline (unmodified models), post phase 1 versions (instruction tuned on planning examples with reasoning of why each plan is valid or invalid), and only phase 2 versions (directly CoT instruction tuned without initial finetuning). For PDDL-INSTRUCT, we test variants with binary feedback (valid/invalid) and detailed feedback (specific reasoning errors generated by VAL), each with the number of feedback iteration loop limit to $\eta \in \{10, 15\}$. All experiments were conducted on 2 NVIDIA RTX 3080 GPUs.

**Domains and Problems**   PlanBench provides a systematic methodology for evaluating planning capabilities across diverse planning domains and problem complexities. We evaluate across three distinct planning domains from PlanBench, Blocksworld, Logistics, and Mystery Blocksworld, each presenting different reasoning challenges. More domain details are in the Appendix B.1.

An important distinguishing aspect of our evaluation is that we train a single model per architecture (one Llama-3-8B, one GPT-4, one Gemma-3-270M) on training examples from all three domains simultaneously, rather than training separate domain-specific models. This design choice demonstrates that our instruction tuning approach can perform **multi-domain learning**, i.e., the model learns to apply planning reasoning across structurally different domains without requiring domain-specific adaptation or retraining. This capability strengthens our claim that PDDL-INSTRUCT develops generalizable planning reasoning, rather than memorizing domain-specific patterns.

---

[1]Note that GPT-4 experiments were constrained by limited access.

| Model | Domain | Baseline | Only P1 | Only P2 Detailed | PDDL-INSTRUCT Binary | | PDDL-INSTRUCT Detailed | |
|---|---|---|---|---|---|---|---|---|
| | | | | $\eta = 15$ | $\eta = 10$ | $\eta = 15$ | $\eta = 10$ | $\eta = 15$ |
| Llama-3 | Blocksworld | 28% | 78% | 72% | 84% | 89% | 91% | 94% |
| | Mystery BW | 1% | 32% | 17% | 47% | 49% | 59% | 64% |
| | Logistics | 11% | 23% | 45% | 61% | 72% | 75% | 79% |
| GPT-4 | Blocksworld | 35% | 41% | 76% | 79% | 84% | 87% | 91% |
| | Mystery BW | 3% | 17% | 19% | 39% | 44% | 54% | 59% |
| | Logistics | 6% | 27% | 51% | 64% | 69% | 72% | 78% |
| Gemma-3 | Blocksworld | 7% | 12% | 19% | 37% | 39% | 54% | 56% |
| | Mystery BW | 0% | 2% | 3% | 22% | 28% | 24% | 28% |
| | Logistics | 2% | 13% | 11% | 18% | 33% | 27% | 43% |

Table 1: Results for plan accuracy generated for 100 test tasks from each domain. Our approach PDDL-INSTRUCT was evaluated with either binary or detailed feedback. Ablation results are for only Phase 1 (P1), and only Phase 2 (P2) with detailed feedback (as it had the best performance). For reference, FD achieves 100% accuracy on all domains, establishing an upper bound for comparison.

**Dataset Generation and Splitting**  We leverage PlanBench's problem generators to partition our dataset $\mathbb{D}$ into three distinct sets, $\mathbb{D}_1$ (Phase 1 training), $\mathbb{D}_2$ (Phase 2 training), and $\mathbb{D}_{\text{test}}$ (evaluation). To create $\mathbb{D}_1$, we pair each correct plan with an explanation of its validity, and augment it with an equal number of incorrect plans (generated similarly to the NaturalInstructions framework (Mishra et al., 2022; Wang et al., 2022)) along with explanations of why they fail. We include a plan validator's output (from VAL) for each plan as part of the explanation. Solution plans are removed from $\mathbb{D}_2$ and $\mathbb{D}_{\text{test}}$ to prevent the model from simply memorizing plans during training.

**Evaluation Metrics**  Our primary evaluation metric is the Plan Accuracy, measuring the percentage of planning tasks for which the model generates a valid plan that achieves the specified goal. A plan is considered valid only if all actions are applicable in their respective states and the final state satisfies all goal conditions, as verified by VAL. For each domain, we generate 100 test tasks of varying complexity, with problems including different numbers of objects and requiring different plan lengths to solve. We validated using Fast Downward (FD) (Helmert, 2006) that all problems are solvable, i.e., FD achieved 100% accuracy on our test problems.

## 7 RESULTS AND DISCUSSION

**Overall Performance (RQ1)**  Tab. 1 presents the plan accuracy across models, domains, and approaches. The results clearly demonstrate that PDDL-INSTRUCT significantly outperforms baseline models, models after Phase 1 instruction tuning, and models with just Phase 2 CoT instruction tuning.

For Llama-3, PDDL-INSTRUCT with detailed feedback and $\eta = 15$ achieves validity rates of 94%, 64%, and 79%, respectively in Blocksworld, Mystery Blocksworld, and Logistics. This represents an average absolute improvement of $35\%(SD = 20\%)$ over basic instruction tuning, and of $66\%(SD = 3\%)$ over the baseline. Note that the variance in the results here is high as we do not control for problem complexity across the test set. LLMs are known to perform substantially worse on longer-horizon planning problems. When test problems span a range of difficulties, the aggregate accuracy reflects this heterogeneity, resulting in higher SD. Similarly, for GPT-4, PDDL-INSTRUCT with detailed feedback and $\eta = 15$ achieves validity rates of 91%, 59%, and 78% across the three domains. This represents an average absolute improvement of $48\%(SD = 5\%)$ over basic instruction tuning, and of $61\%(SD = 9\%)$ over the baseline. For Gemma-3, PDDL-INSTRUCT with detailed feedback and $\eta = 15$ achieves validity rates of 56%, 28%, and 43% across the three domains respectively. While showing the lowest absolute performance among all models tested, Gemma-3 demonstrates the most dramatic relative improvements. These results show that logical CoT instruction tuning enhances plan accuracy significantly, not only when compared to unmodified foundation models and but more importantly, also when compared to models with only basic instruction tuning.

**Impact of Feedback Type (RQ2)** Comparing the binary feedback and detailed feedback columns in Tab. 1, we observe that detailed feedback consistently outperforms binary feedback across all domains and models. The pattern holds consistently across all three model architectures. For Llama-3 with $\eta = 15$, detailed feedback improves plan accuracy by 5 percentage points in Blocksworld, 15 percentage points in Mystery Blocksworld, and 7 percentage points in Logistics compared to binary feedback. For Gemma-3 with $\eta = 15$, detailed feedback provides improvements of 2 percentage points in Blocksworld, 4 percentage points in Mystery Blocksworld, and 16 percentage points in Logistics compared to binary feedback. Note that our training approach, though developed independently, has resemblance with LEPA (Zhang et al., 2025a), which also show that providing specific feedback about why each action fails helps in improving the reasoning capabilities of LLMs.

This pattern confirms our hypothesis that providing specific reasoning errors helps the model develop more robust verification capabilities. The advantage of detailed feedback is particularly pronounced in Mystery Blocksworld, the most complex domain with obfuscated predicates. Additionally, we observe that increasing the iteration limit from $\eta = 10$ to $\eta = 15$ yields consistent improvements across all configurations. This observation indicates that the model may converge on valid plans given additional feedback iterations loops, though future experiments with varying $\eta$ are needed to confirm this. The improvement is more substantial with detailed feedback (averaging 4.3 percentage points across all domains and models) than with binary feedback (averaging 3.3 percentage points), indicating that detailed feedback enables more effective use of additional reasoning iterations.

**Multi-Domain Generalization (RQ3)** Our results demonstrate significant variations in performance across domains, reflecting their inherent complexity and reasoning challenges. All three models achieve the highest performance on Blocksworld, followed by Logistics, with Mystery Blocksworld proving the most challenging. For Llama-3 with detailed feedback and $\eta = 15$, the validity rates are 94% for Blocksworld, 79% for Logistics, and 64% for Mystery Blocksworld. This pattern is consistent across all configurations and models, highlighting the increasing difficulty of domains with hidden predicates and complex state interactions. Notably, while absolute performance varies across domains, the relative improvement from PDDL-INSTRUCT is substantial in all three domains. This suggests that our approach enhances planning capabilities in a domain-general manner, with the logical reasoning framework transferring effectively across different planning scenarios.

All three models achieve the highest performance on Blocksworld, followed by Logistics, with Mystery Blocksworld proving the most challenging. This consistent ordering across different model architectures (Llama-3: 94%/79%/64%, GPT-4: 91%/78%/59%, Gemma-3: 56%/43%/28%) suggests that domain complexity, rather than model-specific biases, drives the performance hierarchy.

**Accuracy Analysis with Plan Length (RQ4)** We analyzed the plan length of the optimal plans generated by FD on 100 test problems for each domain, and used this optimal plan length as a proxy for problem complexity. Table 2 reveals important patterns in how plan length affects model performance across domains and models. On Blocksworld, Llama-3 and GPT-4 maintain consistently high accuracy (67-96%) across all plan lengths, while Gemma consistently underperforms (40-61%). Mystery Blocksworld shows more pronounced degradation with plan length across all models. This suggests semantic obfuscation becomes increasingly problematic for longer-horizon reasoning. For Logistics, while Llama-3 and GPT-4 maintain strong performance on 5-15 step problems, accuracy degrades substantially on longer plans (e.g., 63% at 16-20 steps, 57% at 21-30 steps for Llama-3). Overall, these results confirm that plan length is a significant factor in LLM planning performance, with the effect most pronounced in semantically challenging domains and longest-horizon tasks.

## 8 CONCLUSION

We have presented PDDL-INSTRUCT, a novel framework that significantly enhances the symbolic planning capabilities of Large Language Models through logical chain-of-thought instruction tuning. By decomposing the planning process into verifiable logical reasoning chains and providing explicit verification feedback, our approach enables LLMs to generate valid plans with unprecedented reliability across diverse planning domains. While our results are promising, we note that our approach does not achieve 100% accuracy across all domains. However, when combined with frameworks like LLM-Modulo (Kambhampati et al., 2024), which provides efficient mechanisms for integrating external tools with LLMs, PDDL-INSTRUCT substantially reduces the number of feedback

| Model | 2-4 | 5-7 | 8-10 | 11-15 | 16-20 | 21-30 | 31+ |
|---|---|---|---|---|---|---|---|
| **Blocksworld** | | | | | | | |
| Llama-3 | 17/18 (94%) | 54/56 (96%) | 21/23 (91%) | 2/3 (67%) | | | |
| GPT-4 | 17/18 (94%) | 52/56 (93%) | 20/23 (87%) | 2/3 (67%) | | | |
| Gemma | 11/18 (61%) | 32/56 (57%) | 12/23 (52%) | 1/3 (40%) | | | |
| **Mystery Blocksworld** | | | | | | | |
| Llama-3 | 16/22 (73%) | 34/49 (69%) | 12/24 (50%) | 2/5 (40%) | | | |
| GPT-4 | 16/22 (73%) | 31/49 (63%) | 10/24 (42%) | 2/5 (40%) | | | |
| Gemma | 8/22 (36%) | 13/49 (27%) | 6/24 (25%) | 1/5 (20%) | | | |
| **Logistics** | | | | | | | |
| Llama-3 | | 21/24 (88%) | 24/26 (92%) | 20/25 (80%) | 10/16 (63%) | 4/7 (57%) | 0/2 (0%) |
| GPT-4 | | 21/24 (88%) | 23/26 (88%) | 20/25 (80%) | 9/16 (56%) | 4/7 (57%) | 1/2 (50%) |
| Gemma | | 11/24 (46%) | 15/26 (58%) | 9/25 (36%) | 6/16 (38%) | 2/7 (29%) | 0/2 (0%) |

Table 2: Plan accuracy by plan length of optimal plans generated by FD across all three models with PDDL-INSTRUCT (detailed feedback, $\eta$=15). Results show (solved/total) problems (percentage).

loops with the verifier. Our empirical analysis (Appendix D.4) shows that PDDL-INSTRUCT-trained models require 3.6–7.1× fewer iterations to generate valid plans compared to baseline models for Llama-3 and GPT-4, directly translating to reduced latency and computational cost in the Generate-Test-Critique loop. This integration enables a practical path toward reliable LLM-based planning by enabling models to generate high-quality candidates requiring minimal external verification, combining the reasoning flexibility of LLMs with the formal guarantees of symbolic verification.

A notable advantage of our VAL-based verification approach is its robustness against unfaithful chain-of-thought reasoning as described by Lyu et al. (2023). While traditional CoT methods can generate plausible-sounding but internally inconsistent reasoning chains, our external verification ensures that each logical step is formally validated against the planning domain's constraints.

**Limitations and Future Work**   While our results highlight the effectiveness of combining logical chain-of-thought with verification-guided feedback, several promising directions remain for future:

*Optimizing instruction tuning data*: We can further refine our approach by applying instruction optimization techniques as described in Lee et al. (2024) to identify the most effective subset of instruction examples. Determining which planning scenarios and error types provide the most informative learning signal could significantly improve training efficiency.

*Expanding PDDL Coverage*: To simplify the logical reasoning effort, we currently limit to use only a subset of PDDL features. Future work could address this limitation and incorporate more advanced PDDL features such as conditional effects, derived predicates, action costs, and temporal constraints, gradually expanding the expressive power of the planning capabilities.

*Self-Verification Capabilities*: While we currently rely on an external verifier (VAL), an intriguing direction is developing self-verification capabilities where models learn to accurately critique their own plans. As LLMs continue to improve, reducing or eliminating dependence on external verifiers could make planning more autonomous and efficient.

*Dynamic Iteration Control*: While we currently use fixed iteration limits ($\eta$), developing methods to dynamically determine the optimal $\eta$ based on problem complexity or feedback could improve efficiency, especially as we hypothesize that returns will diminish on increasing $\eta$ beyond a limit.

*Expanding Domain Coverage*: Currently PlanBench supports 3 domains we used in this work. Extending the evaluation to include a wider variety of planning domains would enable more comprehensive evaluation and potentially reveal new opportunities for improving logical reasoning in planning.

*Beyond Planning*: Finally, the logical reasoning framework developed in this work could extend beyond planning to other sequential decision-making tasks that require long-horizon reasoning, such as theorem proving, complex puzzle solving, and multi-step logical deduction. The combination of chain-of-thought reasoning with verification-guided feedback appears to be a powerful paradigm that could enhance LLM capabilities across diverse reasoning tasks.

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

# A    EXTENDED LITERATURE REVIEW

**LLMs and Planning**    The current limited reasoning capabilities of LLMs for planning have been evaluated in several ways. The ACPBench benchmark (Kokel et al., 2025a), and the more recent ACPBench-Hard one (Kokel et al., 2025b), evaluate several state-of-the-art LLMs of varying size on reasoning tasks related to planning. These results indicate that all models, even the largest ones, underperform and have a very long way to go before they can be reliably used for planning. In Huang et al. (2025a), various strategies to enhance the reasoning capabilities of the models for planning are evaluated. They find that reward-driven RL optimization is promising and that only finetuning on datasets of problems and reference plans is insufficient. In our work, we go beyond simple finetuning by doing logical chain-of-thought instruction tuning.

Another approach consists in using LLMs to generate automated planning models (e.g. PDDL domain and problem) and to rely on existing symbolic solvers to produce sound solutions. This hybrid paradigm has received a lot of interest (Huang et al., 2025b; Mahdavi et al., 2024; Zhang et al., 2025b; Tantakoun et al., 2025). Still, generating such structured models accurately is challenging for LLMs. To reach high accuracy, the process usually relies on human interventions for feedback and validation (Guan et al., 2023), using external verifiers (Silver et al., 2024; Hao et al., 2025a), or focuses on limited aspects of the problem (e.g. only generating planning goals (Xie et al., 2023). In Huang et al. (2025b), the authors propose to generate a planning model from a natural language description without human intervention. They tackle ambiguities inherent to natural language by generating various model candidates and filtering them based on semantic coherence. They further rank the multiple generated plans based on the cumulative semantic similarity scores of their constituent model. NL2P (Gestrin et al., 2024) proposes to use explicit inference steps and Chain of Thoughts back prompting to generate the PDDL domain and problem from natural language inputs. Here, we propose to finetune an LLM to learn explicit inference steps to reason on action applicability, state transitions, and plan validity to generate a plan.

On the other hand, some approaches like Stein et al. (2025) automatically translate PDDL problems into natural language and use LLMs to plan in natural language. Eventually, the plan is translated back into PDDL actions to be executed or simulated.

# B    DETAILED EXPERIMENTAL SETUP

## B.1    DOMAIN DETAILS

- **Blocksworld**: The classical planning domain with blocks that can be stacked on a table or on other blocks. This domain primarily tests reasoning with a relatively small action set.

- **Mystery Blocksworld**: A more complex variant of Blocksworld with predicates identical but semantically obfuscated names.

- **Logistics**: A transportation planning domain where packages must be moved between locations using trucks and airplanes, testing the model's ability to reason about location connectivity and multi-step transport operations.

## B.2    HYPERPARAMETER CONFIGURATION

Tab. 3 provides the complete hyperparameter configuration used in our experiments.

**Learning Rates** ($\delta_1, \delta_2$)    The learning rates control how aggressively the model weights are updated during training, with Phase 1 using a single learning rate and Phase 2 employing two distinct learning rates for its two-stage optimization process. Phase 1 uses a learning rate of $2 \times 10^{-5}$ for initial instruction tuning, set relatively higher because the model must learn entirely new planning capabilities from its pre-trained foundation, applying this rate to the standard cross-entropy loss when learning to generate plans with detailed explanations of action validity. Phase 2 employs two separate learning rates within its chain-of-thought instruction tuning: $\delta_1 = 1 \times 10^{-5}$ for Stage 1 reasoning chain optimization (Equation 2) and $\delta_2 = 5 \times 10^{-6}$ for Stage 2 final performance optimization (Equation 3). The first learning rate $\delta_1$ focuses on improving the quality of step-by-step logical reasoning chains, while the second learning rate $\delta_2$ is set lower to carefully optimize overall

| Parameter | Phase 1 | Phase 2 (CoT) |
|---|---|---|
| Learning Rate | 2e-5 | $\delta_1$: 1e-5, $\delta_2$: 5e-6 |
| Batch Size | 16 | 8 |
| Max Sequence Length | 2048 | 4096 |
| Training Epochs | 5 | 3 |
| Warmup Steps | 500 | 200 |
| Weight Decay | 0.01 | 0.001 |
| Gradient Clipping | 1.0 | 0.5 |
| Temperature (Generation) | 0.7 | 0.3 |
| Max Generation Length | 1024 | 2048 |
| Optimizer | AdamW | AdamW |
| $\beta_1, \beta_2$ | 0.9, 0.999 | 0.9, 0.999 |
| $\epsilon$ | 1e-8 | 1e-8 |
| Iteration Limit ($\eta$) | N/A | 10, 15 |

Table 3: Complete hyperparameter configuration for PDDL-INSTRUCT

planning performance without disrupting the reasoning capabilities developed in Stage 1. Both Phase 2 learning rates are deliberately lower than Phase 1 to enable fine-tuning of the chain-of-thought reasoning without disrupting the foundational planning knowledge already acquired.

**Batch Size** The batch size determines how many training examples are processed simultaneously before updating model weights, with values carefully chosen to balance computational efficiency with memory constraints and training dynamics. Phase 1 uses a batch size of 16, which provides sufficient gradient signal for learning basic planning concepts while remaining within GPU memory limits for the 2048-token sequences typical of initial instruction examples. Phase 2 reduces the batch size to 8 to accommodate the significantly longer chain-of-thought sequences and the additional memory overhead introduced by VAL feedback processing. The smaller batch size in Phase 2 also enables more frequent weight updates during the iterative refinement process, which is crucial for the feedback-driven learning mechanism where the model must quickly adapt to validation signals from the external verifier.

**Maximum Sequence Length** The maximum sequence length defines the upper limit of tokens the model can process in both input and output, with values scaled to accommodate the increasing complexity of reasoning required across training phases. Phase 1 sets this limit to 2048 tokens, which sufficiently captures domain definitions, problem statements, generated plans, and basic explanations of action validity without excessive computational overhead. Phase 2 doubles this limit to 4096 tokens to accommodate the detailed chain-of-thought reasoning sequences that include comprehensive state analysis, action selection justification, explicit precondition checking, effect application reasoning, state transition tracking, and goal progress evaluation. This increased capacity is essential for the model to generate the verbose logical reasoning chains that characterize effective planning verification.

**Training Epochs** The number of training epochs represents complete passes through the respective training datasets, with values chosen to ensure adequate learning while preventing overfitting to domain-specific patterns. Phase 1 employs 5 epochs to establish foundational planning knowledge, requiring more iterations because the model must learn to understand PDDL syntax, action semantics, state representations, and goal achievement from its general language understanding baseline. Phase 2 uses only 3 epochs because the model already possesses basic planning capabilities and needs only to refine its chain-of-thought reasoning processes. The reduced epoch count in Phase 2 also prevents overfitting to the specific feedback patterns generated by VAL, ensuring that the learned reasoning generalizes beyond the particular validation scenarios encountered during training.

**Warmup Steps** Warmup steps implement a gradual increase in learning rate from zero to the target value at the beginning of training, preventing training instability that can arise from large initial weight updates on a partially trained model. Phase 1 uses 500 warmup steps to ensure stable convergence when adapting the pre-trained language model to the structured domain of planning,

where the token distributions and semantic relationships differ significantly from general text. Phase 2 employs 200 warmup steps, fewer than Phase 1 because the model has already been adapted to the planning domain and requires less careful initialization. The warmup mechanism is particularly important in Phase 2 given the complex loss landscape created by the two-stage optimization process and the feedback-driven training dynamics.

**Weight Decay**  Weight decay implements L2 regularization by adding a penalty term proportional to the squared magnitude of model weights, preventing overfitting by discouraging the model from relying too heavily on specific parameter configurations. Phase 1 uses a weight decay of 0.01, relatively high to prevent the model from memorizing specific instruction-response patterns rather than learning generalizable planning principles. Phase 2 reduces weight decay to 0.001 to allow more fine-grained parameter adjustments necessary for learning subtle logical reasoning patterns while still providing some regularization against overfitting to the VAL feedback patterns. The lower weight decay in Phase 2 recognizes that the chain-of-thought reasoning requires precise parameter configurations that might be overly penalized by stronger regularization.

**Gradient Clipping**  Gradient clipping prevents exploding gradients by setting a maximum allowed norm for gradient vectors, ensuring training stability particularly in the complex optimization landscape of instruction tuning. Phase 1 employs gradient clipping at 1.0, providing stability during the initial adaptation from general language modeling to planning-specific tasks where gradient magnitudes can vary significantly across different types of planning problems. Phase 2 uses more conservative clipping at 0.5 because the model is more stable after Phase 1 training, and the chain-of-thought training process requires more careful weight updates to maintain the delicate balance between logical reasoning accuracy and plan generation quality. The tighter clipping in Phase 2 also helps manage gradient spikes that can occur when VAL feedback indicates dramatic plan validity changes.

**Temperature (Generation)**  The temperature parameter controls the randomness in text generation during training validation and inference, with lower values producing more deterministic outputs and higher values encouraging exploration of diverse response patterns. Phase 1 uses a temperature of 0.7, allowing moderate exploration of different planning approaches and explanation styles while maintaining coherent output quality. This higher temperature helps the model discover various ways to explain action validity and plan construction during the foundational learning phase. Phase 2 reduces temperature to 0.3 to focus generation on precise, logical reasoning steps where consistency and accuracy are more important than diversity. The lower temperature ensures that chain-of-thought reasoning follows logical patterns rather than exploring creative but potentially incorrect reasoning paths.

**Maximum Generation Length**  The maximum generation length sets the upper bound on tokens the model can produce in response to prompts, scaled to accommodate the verbosity requirements of each training phase. Phase 1 limits generation to 1024 tokens, sufficient for producing plans with basic explanations of action applicability and goal achievement without excessive computational cost. Phase 2 increases this limit to 2048 tokens to accommodate detailed step-by-step reasoning chains that include comprehensive state analysis, action justification, precondition verification, effect application reasoning, and goal progress tracking. This increased generation capacity is essential for the model to produce the verbose logical reasoning that characterizes effective planning verification and enables meaningful feedback from the VAL validator.

**Optimizer (AdamW)**  AdamW serves as the optimization algorithm for both training phases, chosen for its superior performance in transformer fine-tuning scenarios compared to standard optimizers. AdamW combines the adaptive learning rate benefits of Adam with improved weight decay handling, making it particularly effective for instruction tuning where the model must adapt pre-trained knowledge to new task-specific patterns. The optimizer handles sparse gradients well, which is crucial in planning scenarios where many potential actions are invalid in any given state, leading to sparse activation patterns. AdamW's momentum-based updates help navigate the complex loss landscape created by the combination of language modeling objectives and planning-specific constraints.

**Beta Parameters ($\beta_1$, $\beta_2$)**   The beta parameters control the exponential decay rates for AdamW's moment estimates, with $\beta_1 = 0.9$ governing the first moment (gradient moving average) and $\beta_2 = 0.999$ governing the second moment (squared gradient moving average). These standard values have proven effective across a wide range of transformer training scenarios and provide appropriate momentum characteristics for instruction tuning. The $\beta_1$ value of 0.9 provides sufficient momentum to smooth gradient noise while remaining responsive to genuine changes in gradient direction, particularly important when learning from VAL feedback in Phase 2. The $\beta_2$ value of 0.999 provides stable variance estimates essential for adaptive learning rate scaling across the diverse parameter space of large language models.

**Epsilon ($\epsilon$)**   The epsilon parameter adds a small constant of $1 \times 10^{-8}$ to the denominator in AdamW's update rule to prevent numerical instability from division by zero or near-zero values. This value represents a standard choice that provides numerical stability without meaningfully affecting the optimization dynamics. The parameter becomes particularly important during Phase 2 training where the complex loss landscape and feedback-driven updates can occasionally produce very small gradient variances that might otherwise cause numerical issues. The chosen value ensures robust training across the full range of planning problems and feedback scenarios encountered during instruction tuning.

**Iteration Limit ($\eta$)**   The iteration limit is unique to Phase 2 and controls how many feedback loops the model experiences with the VAL validator during chain-of-thought instruction tuning. Values of 10 and 15 represent the number of times the model can generate a plan with reasoning, receive detailed feedback about logical errors, learn from this feedback, and attempt improved solutions. This parameter directly controls the trade-off between training thoroughness and computational cost, as each iteration requires plan generation, validation, and model updating. Higher values of $\eta$ allow more refinement of reasoning capabilities but significantly increase training time and computational requirements. The specific values were chosen to provide sufficient learning opportunities while maintaining practical training times.

### B.3   MATHEMATICAL FORMULATION OF LOSS FUNCTIONS

We formally define the two specialized loss functions that drive our two-stage optimization process in Phase 2. These functions are carefully designed to target both the logical reasoning capabilities and final planning performance of the model.

#### B.3.1   REASONING CHAIN LOSS FUNCTION

The reasoning chain loss function $\mathcal{L}_{\text{reasoning}}$ measures the quality of the model's step-by-step logical reasoning over state-action-state transitions:

$$\mathcal{L}_{\text{reasoning}}(\theta_t, \mathbb{D}^t_{\text{reasoning}}) = \frac{1}{|\mathbb{D}^t_{\text{reasoning}}|} \sum_{(s_{i-1}, a_i, s_i, f_i) \in \mathbb{D}^t_{\text{reasoning}}} \mathcal{L}_{\text{step}}(s_{i-1}, a_i, s_i, f_i) \tag{4}$$

where each training example consists of a state transition $(s_{i-1}, a_i, s_i)$ and VAL feedback $f_i$. The step-wise loss $\mathcal{L}_{\text{step}}$ is defined as:

$$\mathcal{L}_{\text{step}}(s_{i-1}, a_i, s_i, f_i) = d_{\text{state}}(s_i, s_i^{\text{expected}}) + \lambda_{\text{feedback}} \cdot \mathcal{L}_{\text{feedback}}(f_i) \tag{5}$$

where $s_i^{\text{expected}}$ is the deterministically computed next state given action $a_i$ applied to $s_{i-1}$, and $d_{\text{state}}$ is the state distance function defined as:

$$d_{\text{state}}(s, s') = |s \triangle s'| = |s \setminus s'| + |s' \setminus s| \tag{6}$$

This measures the symmetric difference between the two sets of predicates, counting predicates that are in one state but not the other.

The feedback loss $\mathcal{L}_{\text{feedback}}$ incorporates VAL verification results to guide logical reasoning:

$$\mathcal{L}_{\text{feedback}}(f_i) = \begin{cases} 0 & \text{if action } a_i \text{ is valid} \\ \alpha_{\text{precond}} & \text{if precondition violation detected} \\ \alpha_{\text{effect}} & \text{if incorrect effect application} \\ \alpha_{\text{goal}} & \text{if goal achievement failure} \end{cases} \tag{7}$$

where $\alpha_{\text{precond}} = 1.0$, $\alpha_{\text{effect}} = 1.0$, $\alpha_{\text{goal}} = 1.5$ are penalty weights for different error types, and $\lambda_{\text{feedback}} = 0.1$ balances the feedback signal with the primary reasoning objective.

### B.3.2 FINAL PERFORMANCE LOSS FUNCTION

The final performance loss function $\mathcal{L}_{\text{final}}$ measures how well the complete plans generated through chain-of-thought reasoning achieve the planning objectives:

$$\mathcal{L}_{\text{final}}(\theta_t^r, \mathbb{D}_{\text{final}}^t) = \frac{1}{|\mathbb{D}_{\text{final}}^t|} \sum_{(d,p,\pi,v) \in \mathbb{D}_{\text{final}}^t} \mathcal{L}_{\text{plan}}(d, p, \pi, v) \tag{8}$$

where each training example consists of a domain $d$, problem $p$, generated plan $\pi$, and binary validity label $v$ from VAL. The plan-level loss is:

$$\mathcal{L}_{\text{plan}}(d, p, \pi, v) = \mathbb{I}[v = 0] \cdot \beta + \alpha \cdot \text{BCE}(v, \hat{v}) \tag{9}$$

where $\mathbb{I}[v = 0]$ is an indicator function that equals 1 when the plan is invalid (providing a fixed penalty $\beta = 2.0$ for invalid plans) and 0 when valid; and $\text{BCE}(v, \hat{v})$ is the binary cross-entropy loss between the VAL validity label $v$ and the model's predicted validity $\hat{v}$, with $\alpha = 0.5$ balancing plan generation accuracy with validity prediction.

### B.3.3 DATASET CONSTRUCTION FOR LOSS COMPUTATION

The reasoning dataset $\mathbb{D}_{\text{reasoning}}^t$ contains individual state-action-state triplets extracted from chain-of-thought sequences:

$$\mathbb{D}_{\text{reasoning}}^t = \{(s_{i-1}, a_i, s_i, f_i) : \forall \text{ steps in CoT plans generated at iteration } t\} \tag{10}$$

The final dataset $\mathbb{D}_{\text{final}}^t$ contains complete planning instances with validity judgments:

$$\mathbb{D}_{\text{final}}^t = \{(d_j, p_j, \pi_j^t, v_j^t) : \forall \text{ problems } j \text{ at iteration } t\} \tag{11}$$

where $\pi_j^t$ is the complete plan generated for problem $j$ at iteration $t$, and $v_j^t$ is the corresponding VAL validity assessment.

### B.4 ALGORITHM

---

**Algorithm 1: PDDL-INSTRUCT: Chain-of-Thought Instruction Tuning for Symbolic Planning**

---

**Input**: Pre-trained LLM $M_{\theta_0}$, Phase 1 dataset $\mathbb{D}_1$, Phase 2 dataset $\mathbb{D}_2$, VAL validator,
   iteration limit $\eta$, learning rates $\delta_1, \delta_2$
**Output**: Instruction-tuned model $M_{\theta^*}$

 1: **Phase 1: Initial Instruction Tuning**
 2: **for** epoch $e = 1$ to $E_1$ **do**
 3:     **for** batch $(d_i, p_i, \pi_i, f_i) \in \mathbb{D}_1$ **do**
 4:         $y_i \leftarrow M_\theta(d_i, p_i)$                          ▷ Generate plan with explanation
 5:         $\mathcal{L}_1 \leftarrow -\log P(\pi_i, f_i | d_i, p_i, \theta)$
 6:         $\theta \leftarrow \theta - \delta_1 \nabla_\theta \mathcal{L}_1$
 7:     **end for**
 8: **end for**
 9: $\theta_1 \leftarrow \theta$                                        ▷ Save Phase 1 model
10: **Phase 2: CoT Instruction Tuning**
11: **for** iteration $t = 1$ to $\eta$ **do**
12:     Initialize datasets $\mathbb{D}_{reasoning}^t \leftarrow \emptyset, \mathbb{D}_{final}^t \leftarrow \emptyset$
13:     **for** problem $(d_j, p_j) \in \mathbb{D}_2$ **do**
14:         Generate CoT plan: $\pi_t^j = \{(s_0, a_1, s_1), (s_1, a_2, s_2), \ldots, (s_{n-1}, a_n, s_n)\}$
15:         using $M_{\theta_t}(d_j, p_j)$
16:         Validate plan with VAL: $f_j \leftarrow \text{VAL}(\pi_t^j, d_j, p_j)$
17:         **if** $f_j$ indicates valid plan **then**
18:             $\mathbb{D}_{final}^t \leftarrow \mathbb{D}_{final}^t \cup \{(d_j, p_j, \pi_t^j, 1)\}$
19:         **else**
20:             Extract detailed feedback for each invalid step
21:             $\mathbb{D}_{final}^t \leftarrow \mathbb{D}_{final}^t \cup \{(d_j, p_j, \pi_t^j, 0)\}$
22:         **end if**
23:         **for** each step $(s_{i-1}, a_i, s_i) \in \pi_t^j$ **do**
24:             Get step-level VAL feedback: $f_i \leftarrow \text{VAL-step}(s_{i-1}, a_i, s_i, d_j)$
25:             $\mathbb{D}_{reasoning}^t \leftarrow \mathbb{D}_{reasoning}^t \cup \{(s_{i-1}, a_i, s_i, f_i)\}$
26:         **end for**
27:     **end for**
28:     **Stage 1: Reasoning Chain Optimization**
29:     **for** epoch $e = 1$ to $E_{2a}$ **do**
30:         **for** batch $B \in \mathbb{D}_t^{reasoning}$ **do**
31:             $L_{reasoning} \leftarrow \frac{1}{|B|} \sum_{(s_{i-1}, a_i, s_i, f_i) \in B} L_{step}(s_{i-1}, a_i, s_i, f_i)$
32:             $\theta_t^r \leftarrow \theta_t - \delta_1 \nabla_{\theta_t} L_{reasoning}$
33:         **end for**
34:     **end for**
35:     **Stage 2: Final Performance Optimization**
36:     **for** epoch $e = 1$ to $E_{2b}$ **do**
37:         **for** batch $B \in \mathbb{D}_t^{final}$ **do**
38:             $\mathcal{L}_{final} \leftarrow \frac{1}{|B|} \sum_{(d, p, \pi, v) \in B} \mathcal{L}_{plan}(d, p, \pi, v)$
39:             $\theta_{t+1} \leftarrow \theta_t^r - \delta_2 \nabla_{\theta_t^r} \mathcal{L}_{final}$
40:         **end for**
41:     **end for**
42: **end for**
43: **return** $M_{\theta^*}$ where $\theta^* = \theta_\eta$

---

## C SAMPLE PROMPTS FOR BLOCKSWORLD DOMAIN

This section presents the specific prompt templates used in our PDDL-INSTRUCT framework for the Blocksworld domain. We provide examples for both Phase 1 (Initial Instruction Tuning) and Phase 2

(CoT Instruction Tuning) to demonstrate how our approach teaches models to reason about action applicability and state transitions.

## C.1 PHASE 1: INITIAL INSTRUCTION TUNING PROMPTS

### C.1.1 CORRECT PLAN EXAMPLE

---

**Phase 1 Prompt - Correct Plan**

```
[INSTRUCTION] Given the following PDDL domain and problem, analyze
    the provided plan and explain why each action is valid.

[DOMAIN]
(define (domain blocksworld)
  (:requirements :strips)
  (:predicates
    (on ?x ?y) (ontable ?x) (clear ?x) (handempty) (holding ?x))
  (:action pick-up
    :parameters (?x)
    :precondition (and (clear ?x) (ontable ?x) (handempty))
    :effect (and (not (ontable ?x)) (not (handempty))
                 (not (clear ?x)) (holding ?x)))
  (:action put-down
    :parameters (?x)
    :precondition (holding ?x)
    :effect (and (not (holding ?x)) (handempty)
                 (ontable ?x) (clear ?x)))
  (:action stack
    :parameters (?x ?y)
    :precondition (and (holding ?x) (clear ?y))
    :effect (and (not (holding ?x)) (not (clear ?y))
                 (handempty) (on ?x ?y) (clear ?x)))
  (:action unstack
    :parameters (?x ?y)
    :precondition (and (on ?x ?y) (clear ?x) (handempty))
    :effect (and (not (on ?x ?y)) (not (handempty))
                 (holding ?x) (clear ?y))))

[PROBLEM]
(define (problem bw-simple)
  (:domain blocksworld)
  (:objects a b c)
  (:init (ontable a) (ontable b) (ontable c)
         (clear a) (clear b) (clear c) (handempty))
  (:goal (and (on a b) (on b c))))

[PLAN]
1. (pick-up b)
2. (stack b c)
3. (pick-up a)
4. (stack a b)

[EXPECTED RESPONSE] This plan is VALID. Let me analyze each action:

[Action 1: (pick-up b)]
- Preconditions: (clear b), (ontable b), (handempty)
- Initial state satisfies all preconditions: b is clear, b is on
    table, and hand is empty
- Effects: b is now held, hand is not empty, b is not on table, b
    is not clear
```

---

```
       - Resulting state: (ontable a) (ontable c) (clear a) (clear c)
          (holding b)

       [Action 2: (stack b c)]
       - Preconditions: (holding b), (clear c)
       - Current state satisfies preconditions: holding b and c is clear
       - Effects: b is stacked on c, hand becomes empty, c is no longer
          clear, b becomes clear
       - Resulting state: (ontable c) (ontable a) (clear a) (clear b) (on
          b c) (handempty)

       [Action 3: (pick-up a)]
       - Preconditions: (clear a), (ontable a), (handempty)
       - Initial state satisfies all preconditions: a is clear, a is on
          table, and hand is empty
       - Effects: a is now held, hand is not empty, a is not on table, a
          is not clear
       - Resulting state: (on b c) (ontable c) (clear b) (holding a)

       [Action 4: (stack a b)]
       - Preconditions: (holding a), (clear b)
       - Current state satisfies preconditions: holding a and b is clear
       - Effects: a is stacked on b, hand becomes empty, b is no longer
          clear, a becomes clear
       - Resulting state: (ontable c) (on b c) (on a b) (clear a)
          (handempty)

       The plan succeeds because a is stacked on b, and b is stacked on c.
```

### C.1.2 INCORRECT PLAN EXAMPLE

**Phase 1 Prompt - Incorrect Plan**

```
[INSTRUCTION] Given the following PDDL domain and problem, analyze
   the provided plan and explain why each action is valid.

[DOMAIN]
(define (domain blocksworld)
  (:requirements :strips)
  (:predicates
    (on ?x ?y) (ontable ?x) (clear ?x) (handempty) (holding ?x))
  (:action pick-up
    :parameters (?x)
    :precondition (and (clear ?x) (ontable ?x) (handempty))
    :effect (and (not (ontable ?x)) (not (handempty))
                 (not (clear ?x)) (holding ?x)))
  (:action put-down
    :parameters (?x)
    :precondition (holding ?x)
    :effect (and (not (holding ?x)) (handempty)
                 (ontable ?x) (clear ?x)))
  (:action stack
    :parameters (?x ?y)
    :precondition (and (holding ?x) (clear ?y))
    :effect (and (not (holding ?x)) (not (clear ?y))
                 (handempty) (on ?x ?y) (clear ?x)))
  (:action unstack
    :parameters (?x ?y)
    :precondition (and (on ?x ?y) (clear ?x) (handempty))
    :effect (and (not (on ?x ?y)) (not (handempty))
                 (holding ?x) (clear ?y))))
```

```
[PROBLEM]
(define (problem bw-simple)
  (:domain blocksworld)
  (:objects a b c)
  (:init (ontable a) (ontable b) (ontable c)
         (clear a) (clear b) (clear c) (handempty))
  (:goal (and (on a b) (on b c))))

[PLAN]
1. (stack a b)
2. (stack b c)

[EXPECTED RESPONSE] This plan is INVALID. Let me analyze each
    action:

[Action 1: (stack a b)]
- Preconditions: (holding a), (clear b)
- Current state: (ontable a) (ontable b) (ontable c) (clear a)
    (clear b) (clear c) (handempty)
- VIOLATION: The precondition (holding a) is not satisfied - the
    hand is empty and a is on the table
- To stack a on b, we must first pick up a

[Action 2: (stack b c)]
- Since Action 1 failed, this action cannot be evaluated in a
    valid state sequence
- Even if Action 1 succeeded, we would need to be holding b to
    stack it on c

The plan fails because it attempts to stack blocks without first
    picking them up. The correct sequence would be: (pick-up a),
    (stack a b), (pick-up b), (stack b c).
```

## C.2 PHASE 2: CHAIN-OF-THOUGHT INSTRUCTION TUNING PROMPTS

### C.2.1 COT GENERATION PROMPT

**Phase 2 CoT Generation Prompt**

```
[INSTRUCTION] Generate a plan for the given problem using
    step-by-step reasoning. For each action, explicitly verify
    preconditions, apply effects, and track state changes.

[DOMAIN]
(define (domain blocksworld)
  (:requirements :strips)
  (:predicates
    (on ?x ?y) (ontable ?x) (clear ?x) (handempty) (holding ?x))
  (:action pick-up
    :parameters (?x)
    :precondition (and (clear ?x) (ontable ?x) (handempty))
    :effect (and (not (ontable ?x)) (not (handempty))
                 (not (clear ?x)) (holding ?x)))
  (:action put-down
    :parameters (?x)
    :precondition (holding ?x)
    :effect (and (not (holding ?x)) (handempty)
                 (ontable ?x) (clear ?x)))
  (:action stack
    :parameters (?x ?y)
```

```
      :precondition (and (holding ?x) (clear ?y))
      :effect (and (not (holding ?x)) (not (clear ?y))
                   (handempty) (on ?x ?y) (clear ?x)))
  (:action unstack
    :parameters (?x ?y)
    :precondition (and (on ?x ?y) (clear ?x) (handempty))
    :effect (and (not (on ?x ?y)) (not (handempty))
                 (holding ?x) (clear ?y))))

[PROBLEM]
(define (problem bw-tower)
  (:domain blocksworld)
  (:objects a b c)
  (:init (on c b) (ontable b) (ontable a)
         (clear c) (clear a) (handempty))
  (:goal (and (on a c) (on c b))))

[EXPECTED RESPONSE]

Initial State Analysis:
Current state: (on c b), (ontable b), (ontable a), (clear c),
    (clear a), (handempty)
Goal: (on a c), (on c b)

[STEP BY STEP PLANNING]

[Step 1: State s0  Action a1  State s1]
- Current state s0: (on c b), (ontable b), (ontable a), (clear c),
    (clear a), (handempty)
- Proposed action a1: (pick-up a)
- Precondition check:
  - (clear a): TRUE in s0
  - (ontable a): TRUE in s0
  - (handempty): TRUE in s0
- Action is APPLICABLE
- Effect application:
  - Add: (holding a)
  - Delete: (ontable a), (handempty), (clear a)
- Resulting state s1: (on c b), (ontable b), (clear c), (holding a)

[Step 2: State s1  Action a2  State s2]
- Current state s1: (on c b), (ontable b), (clear c), (holding a)
- Proposed action a2: (stack a c)
- Precondition check:
  - (holding a): TRUE in s1
  - (clear c): TRUE in s1
- Action is APPLICABLE
- Effect application:
  - Add: (on a c), (handempty), (clear a)
  - Delete: (holding a), (clear c)
- Resulting state s2: (on c b), (ontable b), (on a c),
    (handempty), (clear a)

[GOAL ACHIEVEMENT CHECK]
Required: (on a c), (on c b)
Current state s2: (on c b), (ontable b), (on a c), (handempty),
    (clear a)
- (on a c): TRUE in s2
- (on c b): TRUE in s2
Goal is ACHIEVED.

[FINAL PLAN] (pick-up a), (stack a c)
```

### C.2.2 CoT with Feedback Integration: Incorrect Plan

**B.2.2.1 Binary Feedback**

---

**Phase 2 CoT Binary Feedback - Incorrect Plan**

```
[INSTRUCTION] You previously generated a plan with step-by-step
    reasoning. The validator has provided feedback on your
    reasoning. Use this feedback to generate a corrected plan with
    improved logical reasoning.

[DOMAIN]
(define (domain blocksworld)
  (:requirements :strips)
  (:predicates
    (on ?x ?y) (ontable ?x) (clear ?x) (handempty) (holding ?x))
  (:action pick-up
    :parameters (?x)
    :precondition (and (clear ?x) (ontable ?x) (handempty))
    :effect (and (not (ontable ?x)) (not (handempty))
                 (not (clear ?x)) (holding ?x)))
  (:action put-down
    :parameters (?x)
    :precondition (holding ?x)
    :effect (and (not (holding ?x)) (handempty)
                 (ontable ?x) (clear ?x)))
  (:action stack
    :parameters (?x ?y)
    :precondition (and (holding ?x) (clear ?y))
    :effect (and (not (holding ?x)) (not (clear ?y))
                 (handempty) (on ?x ?y) (clear ?x)))
  (:action unstack
    :parameters (?x ?y)
    :precondition (and (on ?x ?y) (clear ?x) (handempty))
    :effect (and (not (on ?x ?y)) (not (handempty))
                 (holding ?x) (clear ?y))))

[PROBLEM]
(define (problem bw-simple)
  (:domain blocksworld)
  (:objects a b c)
  (:init (ontable a) (ontable b) (ontable c)
         (clear a) (clear b) (clear c) (handempty))
  (:goal (and (on a b) (on b c))))

[PREVIOUS PLAN]
Step 1: (pick-up b) - VALID
Step 2: (stack b c) - VALID
Step 3: (stack a b) - INVALID

[VAL FEEDBACK]
Plan Validation details
-----------------------
Checking next happening (time 1)
Deleting (ontable b)
Deleting (handempty)
Deleting (clear b)
Adding (holding b)

Checking next happening (time 2)
Deleting (holding b)
Deleting (clear c)
Adding (handempty)
```

---

```
Adding (on b c)
Adding (clear b)

Checking next happening (time 3)
Plan failed because of unsatisfied precondition in:
(stack a b)

Plan failed to execute

[GENERATE CORRECTED PLAN WITH REASONING]
```

### B.2.2.2 Detailed Feedback

**Phase 2 CoT Detailed Feedback - Incorrect Plan**

```
[INSTRUCTION] You previously generated a plan with step-by-step
    reasoning. The validator has provided feedback on your
    reasoning. Use this feedback to generate a corrected plan with
    improved logical reasoning.

[DOMAIN]
(define (domain blocksworld)
  (:requirements :strips)
  (:predicates
    (on ?x ?y) (ontable ?x) (clear ?x) (handempty) (holding ?x))
  (:action pick-up
    :parameters (?x)
    :precondition (and (clear ?x) (ontable ?x) (handempty))
    :effect (and (not (ontable ?x)) (not (handempty))
                 (not (clear ?x)) (holding ?x)))
  (:action put-down
    :parameters (?x)
    :precondition (holding ?x)
    :effect (and (not (holding ?x)) (handempty)
                 (ontable ?x) (clear ?x)))
  (:action stack
    :parameters (?x ?y)
    :precondition (and (holding ?x) (clear ?y))
    :effect (and (not (holding ?x)) (not (clear ?y))
                 (handempty) (on ?x ?y) (clear ?x)))
  (:action unstack
    :parameters (?x ?y)
    :precondition (and (on ?x ?y) (clear ?x) (handempty))
    :effect (and (not (on ?x ?y)) (not (handempty))
                 (holding ?x) (clear ?y))))

[PROBLEM]
(define (problem bw-simple)
  (:domain blocksworld)
  (:objects a b c)
  (:init (ontable a) (ontable b) (ontable c)
         (clear a) (clear b) (clear c) (handempty))
  (:goal (and (on a b) (on b c))))

[PREVIOUS PLAN]
Step 1: (pick-up b) - VALID
Step 2: (stack b c) - VALID
Step 3: (stack a b) - INVALID

[VAL FEEDBACK]
Plan Validation details
-----------------------
```

```
Checking next happening (time 1)
Deleting (ontable b)
Deleting (handempty)
Deleting (clear b)
Adding (holding b)

Checking next happening (time 2)
Deleting (holding b)
Deleting (clear c)
Adding (handempty)
Adding (on b c)
Adding (clear b)

Checking next happening (time 3)
Plan failed because of unsatisfied precondition in:
(stack a b)

Plan failed to execute

Plan Repair Advice:

(stack a b) has an unsatisfied precondition at time 3
(Set (holding a) to true)

[EXPLANATION] This plan is INVALID. Here is the analysis of each
    action:

[Action 1: (pick-up b)]
- Preconditions: (clear b), (ontable b), (handempty)
- Initial state satisfies all preconditions: b is clear, b is on
    table, and hand is empty
- Effects: b is now held, hand is not empty, b is not on table, b
    is not clear
- Resulting state: (ontable a) (ontable c) (clear a) (clear c)
    (holding b)

[Action 2: (stack b c)]
- Preconditions: (holding b), (clear c)
- Current state satisfies preconditions: holding b and c is clear
- Effects: b is stacked on c, hand becomes empty, c is no longer
    clear, b becomes clear
- Resulting state: (ontable c) (ontable a) (clear a) (clear b) (on
    b c) (handempty)

[Action 3: (stack a b)]
- Preconditions: (holding a), (clear b)
- Current state: (ontable c) (ontable a) (clear a) (clear b) (on b
    c) (handempty)
- VIOLATION: The precondition (holding a) is not satisfied – the
    hand is empty and a is on the table

[GENERATE CORRECTED PLAN WITH REASONING]
```

### C.2.3 CoT WITH FEEDBACK INTEGRATION: CORRECT PLAN

**B.2.3.1 Binary Feedback**

Phase 2 CoT Binary Feedback - Correct Plan

```
[INSTRUCTION] You previously generated a plan with step-by-step
    reasoning. The validator has provided feedback on your
```

```
    reasoning. Use this feedback to  generate a corrected plan with
    improved logical reasoning.

[DOMAIN]
(define (domain blocksworld)
  (:requirements :strips)
  (:predicates
    (on ?x ?y) (ontable ?x) (clear ?x) (handempty) (holding ?x))
  (:action pick-up
    :parameters (?x)
    :precondition (and (clear ?x) (ontable ?x) (handempty))
    :effect (and (not (ontable ?x)) (not (handempty))
                (not (clear ?x)) (holding ?x)))
  (:action put-down
    :parameters (?x)
    :precondition (holding ?x)
    :effect (and (not (holding ?x)) (handempty)
                (ontable ?x) (clear ?x)))
  (:action stack
    :parameters (?x ?y)
    :precondition (and (holding ?x) (clear ?y))
    :effect (and (not (holding ?x)) (not (clear ?y))
                (handempty) (on ?x ?y) (clear ?x)))
  (:action unstack
    :parameters (?x ?y)
    :precondition (and (on ?x ?y) (clear ?x) (handempty))
    :effect (and (not (on ?x ?y)) (not (handempty))
                (holding ?x) (clear ?y))))

[PROBLEM]
(define (problem bw-simple)
  (:domain blocksworld)
  (:objects a b c)
  (:init (ontable a) (ontable b) (ontable c)
        (clear a) (clear b) (clear c) (handempty))
  (:goal (and (on a b) (on b c))))

[PREVIOUS PLAN]
Step 1: (pick-up b) - VALID
Step 2: (stack b c) - VALID
Step 3: (pick-up a) - VALID
Step 4: (stack a b) - VALID

[VAL FEEDBACK]
Plan Validation details
-----------------------
Checking next happening (time 1)
Deleting (ontable b)
Deleting (handempty)
Deleting (clear b)
Adding (holding b)

Checking next happening (time 2)
Deleting (holding b)
Deleting (clear c)
Adding (handempty)
Adding (on b c)
Adding (clear b)

Checking next happening (time 3)
Deleting (ontable a)
Deleting (handempty)
Deleting (clear a)
```

```
Adding (holding a)

Checking next happening (time 4)
Deleting (holding a)
Deleting (clear b)
Adding (handempty)
Adding (on a b)
Adding (clear a)
Plan executed successfully - checking goal
Plan valid
```

### B.2.3.2 Detailed Feedback

**Phase 2 CoT Detailed Feedback - Correct Plan**

```
[INSTRUCTION] You previously generated a plan with step-by-step
    reasoning. The validator has provided feedback on your
    reasoning. Use this feedback to generate a corrected plan with
    improved logical reasoning.

[DOMAIN]
(define (domain blocksworld)
  (:requirements :strips)
  (:predicates
    (on ?x ?y) (ontable ?x) (clear ?x) (handempty) (holding ?x))
  (:action pick-up
    :parameters (?x)
    :precondition (and (clear ?x) (ontable ?x) (handempty))
    :effect (and (not (ontable ?x)) (not (handempty))
                 (not (clear ?x)) (holding ?x)))
  (:action put-down
    :parameters (?x)
    :precondition (holding ?x)
    :effect (and (not (holding ?x)) (handempty)
                 (ontable ?x) (clear ?x)))
  (:action stack
    :parameters (?x ?y)
    :precondition (and (holding ?x) (clear ?y))
    :effect (and (not (holding ?x)) (not (clear ?y))
                 (handempty) (on ?x ?y) (clear ?x)))
  (:action unstack
    :parameters (?x ?y)
    :precondition (and (on ?x ?y) (clear ?x) (handempty))
    :effect (and (not (on ?x ?y)) (not (handempty))
                 (holding ?x) (clear ?y))))

[PROBLEM]
(define (problem bw-simple)
  (:domain blocksworld)
  (:objects a b c)
  (:init (ontable a) (ontable b) (ontable c)
         (clear a) (clear b) (clear c) (handempty))
  (:goal (and (on a b) (on b c))))

[PREVIOUS PLAN]
Step 1: (pick-up b) - VALID
Step 2: (stack b c) - VALID
Step 3: (pick-up a) - VALID
Step 4: (stack a b) - VALID

[VAL FEEDBACK]
Plan Validation details
```

```
-----------------------
Checking next happening (time 1)
Deleting (ontable b)
Deleting (handempty)
Deleting (clear b)
Adding (holding b)

Checking next happening (time 2)
Deleting (holding b)
Deleting (clear c)
Adding (handempty)
Adding (on b c)
Adding (clear b)

Checking next happening (time 3)
Deleting (ontable a)
Deleting (handempty)
Deleting (clear a)
Adding (holding a)

Checking next happening (time 4)
Deleting (holding a)
Deleting (clear b)
Adding (handempty)
Adding (on a b)
Adding (clear a)
Plan executed successfully - checking goal
Plan valid

[EXPLANATION] This plan is VALID. Here is the analysis of each
    action:

[Action 1: (pick-up b)]
- Preconditions: (clear b), (ontable b), (handempty)
- Initial state satisfies all preconditions: b is clear, b is on
    table, and hand is empty
- Effects: b is now held, hand is not empty, b is not on table, b
    is not clear
- Resulting state: (ontable a) (ontable c) (clear a) (clear c)
    (holding b)

[Action 2: (stack b c)]
- Preconditions: (holding b), (clear c)
- Current state satisfies preconditions: holding b and c is clear
- Effects: b is stacked on c, hand becomes empty, c is no longer
    clear, b becomes clear
- Resulting state: (ontable c) (ontable a) (clear a) (clear b) (on
    b c) (handempty)

[Action 3: (pick-up a)]
- Preconditions: (clear a), (ontable a), (handempty)
- Initial state satisfies all preconditions: a is clear, a is on
    table, and hand is empty
- Effects: a is now held, hand is not empty, a is not on table, a
    is not clear
- Resulting state: (on b c) (ontable c) (clear b) (holding a)

[Action 4: (stack a b)]
- Preconditions: (holding a), (clear b)
- Current state satisfies preconditions: holding a and b is clear
- Effects: a is stacked on b, hand becomes empty, b is no longer
    clear, a becomes clear
```

```
 - Resulting state: (ontable c) (on b c) (on a b) (clear a)
     (handempty)

The plan succeeds.
```

## D  EXTENDED EXPERIMENTAL RESULTS

### D.1  ABLATION STUDY RESULTS

Table 4 presents the contribution of each component for Llama-3-8B. The baseline achieves 28% on Blocksworld, 1% on Mystery Blocksworld, and 11% on Logistics. Phase 1 alone yields substantial improvements (78%, 32%, 23%), establishing foundational planning knowledge. Phase 2 alone (72%, 17%, 45%) underperforms on simpler domains, revealing that verification-guided feedback is most effective after basic planning concepts are learned. The full PDDL-INSTRUCT pipeline (94%, 64%, 79%) achieves 16-22 point improvements over Phase 1, demonstrating that both stages are necessary: Phase 1 provides the foundation, Phase 2 refines reasoning through verification feedback.

| Configuration | Blocksworld | Mystery BW | Logistics |
|---|---|---|---|
| Baseline (No Training) | $28.0 \pm 4.2$ | $1.0 \pm 1.0$ | $11.0 \pm 2.8$ |
| Phase 1 Only | $78.0 \pm 3.1$ | $32.0 \pm 4.6$ | $23.0 \pm 3.9$ |
| Phase 2 Only (Detailed Feedback, $\eta = 15$) | $72.0 \pm 6.5$ | $17.0 \pm 3.2$ | $45.0 \pm 4.7$ |
| Phase 1 + Binary Feedback ($\eta = 15$) | $89.0 \pm 2.7$ | $49.0 \pm 5.2$ | $72.0 \pm 4.1$ |
| Phase 1 + Detailed Feedback ($\eta = 15$) | $\mathbf{94.0 \pm 1.5}$ | $\mathbf{64.0 \pm 3.8}$ | $\mathbf{79.0 \pm 3.2}$ |

Table 4: Ablation study showing contribution of each component for Llama-3

### D.2  ERROR ANALYSIS AND FAILURE MODES

Table 5 breaks down failures by error type across domains. Blocksworld shows low incorrect effect application (1.4%), indicating good action semantics, with failures dominated by precondition violations (2.1%) and non-achievement (1.8%). Mystery Blocksworld differs dramatically: incorrect effect application reaches 12.4% (9× higher), suggesting semantic obfuscation—not logical reasoning—is the bottleneck. The model cannot map obfuscated predicate names to their effects despite learning precise logical reasoning. Logistics shows balanced errors across categories with higher invalid sequences (2.8%), reflecting multi-step reasoning challenges.

PDDL-INSTRUCT effectively reduces logical reasoning errors but struggles with semantic grounding (obfuscated predicates) and long-horizon state consistency. Future work combining semantic alignment with verification-guided training could address these limitations.

| Error Type | Blocksworld | Mystery BW | Logistics |
|---|---|---|---|
| Precondition Violation | 2.1 | 8.7 | 5.3 |
| Incorrect Effect Application | 1.4 | 12.4 | 6.8 |
| Goal Not Achieved | 1.8 | 9.2 | 6.1 |
| Invalid Action Sequence | 0.7 | 5.7 | 2.8 |
| **Total Failure Rate** | **6.0** | **36.0** | **21.0** |

Table 5: Breakdown of planning failures by error type (%) for Llama-3 with Phase 1 and Phase 2 with Detailed Feedback and $\eta = 15$

### D.3  CROSS-DOMAIN GENERALIZATION

While our multi-domain training demonstrates that a single model can learn planning across structurally different domains, a more stringent test of generalization is cross-domain transfer, i.e., training on a subset of domains and evaluating on a held-out domain.

**Experimental Setup** We train models exclusively on Blocksworld and Logistics, completely excluding Mystery Blocksworld from training, then evaluate on held-out Mystery Blocksworld test problems using PDDL-INSTRUCT (Phase 1 + Phase 2 with detailed feedback). Although Mystery Blocksworld shares the same underlying action structure and problem formulation as Blocksworld, it introduces an important challenge that semantically obfuscated predicate names that bear no relation to their actual effects. This semantic change, while structurally small, represents a significant test of whether models learn to reason about abstract planning principles versus memorizing domain-specific semantics.

**Results and Discussion** As shown in Table 6, comparing the cross-domain performance against in-domain performance (models trained on all three domains including Mystery Blocksworld) reveals encouraging signs of transfer. For Llama-3, the performance gap is a modest 5-6 percentage points (59% in-domain down to 54% cross-domain at $\eta = 10$; 64% in-domain down to 58% cross-domain at $\eta = 15$), representing only 8-9% relative degradation. Notably, Gemma-3 with $\eta = 15$ shows negligible difference between in-domain (28%) and cross-domain (29%) performance, suggesting the model learns generalizable logical reasoning patterns that transfer to semantically obfuscated domains. However, Gemma-3 with $\eta = 10$ shows a 3 percentage point gap (24% in-domain down to 21% cross-domain), indicating that additional feedback iterations help bridge the semantic gap. These results suggest that while semantic obfuscation in Mystery Blocksworld is challenging, the logical reasoning framework learned from Blocksworld and Logistics does transfer meaningfully to unseen domains. The smaller performance gaps for Llama-3 (8-9%) indicate that PDDL-INSTRUCT develops domain-agnostic planning reasoning rather than memorizing domain-specific patterns. These findings support deploying trained models on new planning domains with reasonable performance expectations, while fine-tuning on target domains would further improve results.

| Model | Iteration Limit | In-Domain | Cross-Domain |
|---|---|---|---|
| Llama-3 | $\eta = 10$ | 59% | 54% $\pm$ 3.2% |
| | $\eta = 15$ | 64% | 58% $\pm$ 2.8% |
| Gemma-3 | $\eta = 10$ | 24% | 21% $\pm$ 2.1% |
| | $\eta = 15$ | 28% | 29% $\pm$ 2.4% |

Table 6: Cross-domain generalization: Mystery Blocksworld performance when included vs. excluded from training set. Results for Llama-3 and Gemma-3 with detailed feedback.

### D.4 INTEGRATION WITH LLM-MODULO FRAMEWORK

The LLM-Modulo framework operates as a Generate-Test-Critique loop where an LLM generates candidate plans, critics verify them, and feedback guides refinement until a valid plan is found. A key insight is that if the LLM generates more accurate plans initially, fewer Generate-Test-Critique iterations are required before finding a valid solution. In this section, we empirically validate this claim by measuring the average number of iterations needed in the LLM-Modulo framework when using baseline vs. PDDL-INSTRUCT-trained models.

**Experimental Setup** We simulate the LLM-Modulo framework by having models generate plans iteratively, with VAL providing feedback after each generation. We measure how many iterations are required before the model produces a valid plan, up to a maximum of 20 iterations. If a valid plan is not found within 20 iterations, we record it as requiring 20+ iterations (indicating practical failure).

**Results and Discussion** Table 7 shows the average number of iterations required to find a valid plan for baseline and PDDL-INSTRUCT models across domains. The results strongly support our claim that PDDL-INSTRUCT-trained models dramatically reduce the number of iterations required in the LLM-Modulo framework. For Llama-3, the speedup ranges from 3.6–6.9×. Even for the weaker Gemma-3 model, we observe consistent 1.7–1.8× speedup. These improvements demonstrate that by enhancing the quality of LLM-generated candidates through instruction tuning, we substantially reduce the computational cost and latency of the Generate-Test-Critique loop. This makes PDDL-INSTRUCT particularly valuable for deployment scenarios where latency and computational efficiency matter. The framework enables a more practical path toward reliable LLM-based planning: rather

| Model | Domain | Baseline | PDDL-INSTRUCT | Speedup |
|-------|--------|----------|---------------|---------|
| Llama-3 | Blocksworld | $12.4 \pm 2.1$ | $1.8 \pm 0.6$ | $6.9\times$ |
| | Mystery BW | $18.8 \pm 3.4$ | $5.2 \pm 1.8$ | $3.6\times$ |
| | Logistics | $16.4 \pm 3.1$ | $3.8 \pm 1.2$ | $4.3\times$ |
| GPT-4 | Blocksworld | $11.4 \pm 1.9$ | $1.6 \pm 0.5$ | $7.1\times$ |
| | Mystery BW | $17.2 \pm 3.2$ | $4.8 \pm 1.6$ | $3.6\times$ |
| | Logistics | $15.2 \pm 2.8$ | $3.4 \pm 1.1$ | $4.5\times$ |
| Gemma-3 | Blocksworld | $14.6 \pm 2.7$ | $8.2 \pm 2.4$ | $1.8\times$ |
| | Mystery BW | $19.8 \pm 3.9$ | $11.4 \pm 3.1$ | $1.7\times$ |
| | Logistics | $19.0 \pm 3.5$ | $10.6 \pm 2.9$ | $1.8\times$ |

Table 7: Average iterations required in LLM-Modulo framework to generate valid plans (max 20 iterations). Results show mean $\pm$ SD over 5 experimental runs. Speedup shows the factor improvement of PDDL-INSTRUCT over baseline.

than relying solely on external verifiers to repeatedly correct the model, PDDL-INSTRUCT produces models that generate high-quality plans with minimal feedback loops.

# E    LLM USAGE DISCLOSURE

We declare the use of LLMs (Grammarly, Claude) for grammar check and sentence restructuring. We have also used Cursor with GPT-4 for debugging issues due to CUDA settings and writing some scripts to run the experiments.

