In this paper, we ~~challenge this limitation by introducing~~ introduce PDDL-INSTRUCT, a novel framework shown in Fig. 1, that augments LLMs' reasoning capabilities with the formal reasoning required for automated planning. PDDL-INSTRUCT explicitly teaches LLMs to reason through the precondition-effect structure of planning domains using logical chain-of-thought prompting. ~~By decomposing planning verification into atomic reasoning steps and incorporating this structure into instruction tuning, our approach enables LLMs to not only generate syntactically correct plans but also to verify their logical validity through step-by-step reasoning. This ability to perform structured verification significantly enhances the reliability of LLM-generated plans and opens up possibilities for self-correction through iterative refinement~~ Rather than attempting to compete with symbolic planners which achieve 100% accuracy on these problems, our goal is to improve how LLMs learn to reason about planning through systematic instruction tuning and verification feedback. This foundational improvement in LLM planning capabilities is a necessary step toward developing hybrid systems that leverage both LLM flexibility and symbolic verification.

Main contributions of this paper are:

- A ~~novel~~ parameter-update-based instruction tuning framework that enhances symbolic planning ~~capabilities~~ in LLMs through logical chain-of-thought reasoning ~~, focusing specifically on plan generation and action applicability verification~~.

- A formalization of ~~the planning verification process~~ planning verification as decomposable reasoning chains with domain-informed loss functions, enabling LLMs to systematically check preconditions, apply effects, and validate invariants. This allows gradient-based optimization to directly target planning semantics rather than generic language modeling objectives.

- Empirical evidence ~~demonstrating that~~ that parameter-update-based instruction-tuned LLMs ~~can develop robust~~ planning capabilities ~~that generalize across domains~~

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

~~**Data Splitting** As shown in Fig. 1, our approach has three phases (more details in Sec. 5). To facilitate this, we partition the dataset $\mathbb{D}$ into three sets: $\mathbb{D}_1$, $\mathbb{D}_2$, and $\mathbb{D}_{\text{test}}$ for Phase 1 training, Phase 2 training, and evaluation, respectively.~~

~~We add additional data to $\mathbb{D}_1$ by adding incorrect plans for each problem, similar to NaturalInstructions framework (Mishra et al., 2022; Wang et al., 2022). We also add a plan validator's output for each plan (both correct and incorrect) as an explanation to the dataset. We remove the solution plans from datasets $\mathbb{D}_2$ and $\mathbb{D}_{test}$.~~