# OpenReview forum: "Teaching LLMs to Plan: Logical Chain-of-Thought Instruction Tuning for Symbolic Planning"
_ICLR.cc/2026/Conference — Submitted to ICLR 2026_

### Official Review · Reviewer_gPN4 · 2025-10-21

**Soundness:** 3
**Presentation:** 3
**Contribution:** 2
**Rating:** 4
**Confidence:** 3

**Summary:**

The authors propose the PDDL-Instruct framework, a model finetuning approach, to improve Large Language Models abilities on sequential planning tasks. The finetuning is split into two phases. The first phase is an instruction tuning phase, where the model is trained over both correct and incorrect plans. The model output for each plan is an analysis of whether the given plan is valid and why it is or is not valid. The second phase ia a chain-of-though tuningand  phase. The model is prompted to generate a chain of planning tuples, which is checked for correctness by an external verification algorithm. The input to a model tuned by this framework is then a set of natural language instructions, a domain description in PDDL, and goal to be reached described in PDDL. The LLM output is a set of (State, Action, Next State) tuples that can be used to reach the goal.

**Strengths:**

1) The finetuning framework combines both existing strategies and new finetuning approaches
   2)  The writing is clear and the provided tables / figures are easy to understand
    3) Inclusion of error analysis and failure modes is useful for understanding how future frameworks can improve this framework and also for helping interperate the results in the main paper
   4)  Training on negatives is not a new concept, but it is not one I've seen applied in the context of LLM planning, and is an idea I feel is typically neglected. Clever of the athors to include such an approach.
   5)  Results show strong improvements over baseline models
   6) Moderate novelty and significance

**Weaknesses:**

1) The authors do not compare against any other planning frameworks. Even if the other frameworks were not made for this task and the the authors were sure that the other frameworks would fail, it is still useful to see the performance of other frameworks to put the results in context. Can these problems be easily solved by a general solver for AI planning problems?
2) This work has some similarity with the approach in this paper that also uses feedback: https://arxiv.org/abs/2309.16436 However, that paper does not use any fine-tuning.
3) I guess the overall excitement about the paper is rather limited as all the the applied techniques are well established. However, they may not have been applied together to this type of problem.
4) The authors include some figures and tables in the appendices. It would be beneficial if descriptions of the figures/tables and analysis of the results were included.
5) Minimal results were provides. See questions.

**Questions:**

1) Have the authors tried training on one dataset (Blocksworld) and then testing on another (Logistics)? How well does the model generalize to unseen data?
2) Have the authors trained on two of the datasets (Blocksworld, Logistics) and tested on the heldout dataset (Blocksworld Mystery)?

---

> ### Author Response · Authors · 2025-11-24
>
> We thank the reviewer for the thoughtful feedback. We address each concern below:
>
> ## Weakness 1
>
> > Can these problems be easily solved by a general solver for AI planning problems?
>
> Please refer to C1 in the common response
>
> ## Weakness 2 (Relationship to Suggested Prior Work)
>
> > This work has some similarity with the approach in this paper that also uses feedback: https://arxiv.org/abs/2309.16436
>
> Thank you for sharing this reference. On careful reading of the paper, we found the idea similar to existing LLM+verification based planning frameworks that we discuss in Sec. 2 under LLMs for planning. We acknowledge these connections, and discuss them in our common response C2.
>
> ## Weakness 3
>
> > I guess the overall excitement about the paper is rather limited as all the the applied techniques are well established. However, they may not have been applied together to this type of problem.
>
> We acknowledge the reviewer's observation that individual techniques (finetuning, instruction tuning, CoT prompting, external verification) are well-established in isolation. However, the core contribution of this work lies in their systematic integration for LLM-based PDDL planning along with model parameter updates. While each component is independently standard, their combination along with model updates is novel in something no-one has tried before, and our empirical results support the value of this integration. We achieve a 66% absolute improvement over baseline on Blocksworld with Llama-3, consistent improvements across three domains and three model sizes, and nuanced insights from our binary vs. detailed feedback ablation that reveal important effects of feedback design.
>
> ## Weakness 4
>
> > The authors include some figures and tables in the appendices. It would be beneficial if descriptions of the figures/tables and analysis of the results were included.
>
> We have updated the appendix with comprehensive analysis including error breakdown by failure type (Table 4), extended ablation results with standard deviations (Table 3), complete algorithmic description (Algorithm 1), and hyperparameter justifications (Appendix B.1). In the revision, we will add accompanying narrative analysis and interpretation for each table to improve clarity.
>
> ## Weakness 5, Questions 1 and 2
>
> > Have the authors tried training on one dataset (Blocksworld) and then testing on another (Logistics)? How well does the model generalize to unseen data?
> > Have the authors trained on two of the datasets (Blocksworld, Logistics) and tested on the heldout dataset (Blocksworld Mystery)?
>
> Please refer to common response C4

---

### Official Review · Reviewer_arPJ · 2025-10-30

**Soundness:** 3
**Presentation:** 3
**Contribution:** 2
**Rating:** 4
**Confidence:** 4

**Summary:**

The paper proposes PDDL-INSTRUCT, an instruction-tuning framework to improve LLMs’ classical planning in PDDL domains. The central move is to turn plan generation into explicit state–action–state chains and supervise them with a formal validator (VAL) that checks preconditions, effects, and goal satisfaction. Training is staged: first general instruction-tuning with explanations (including negative examples), then a chain-of-thought phase with two objectives: a step-level “reasoning-chain” loss over ⟨si−1, ai, si⟩ triplets and a final plan-validity loss. On PlanBench-style tasks, reported accuracy gains over base models and a Phase-1-only ablation are large, with the strongest results when using detailed validator feedback and more feedback iterations.

The contribution is best viewed as an integrative advance rather than a new paradigm. Each major component has precedent namely: instruction tuning, chain-of-thought prompting, and external verification, but the paper’s value lies in packaging them into a coherent training loop that treats symbolic planning as verifiable, stepwise reasoning and shows strong empirical gains. The two-stage loss that first optimizes step-level reasoning chains and then end-task validity is a nice, concrete design choice; using detailed validator feedback to supervise CoT at the triplet level is what most clearly differentiates this from prior self-critique/reflection approaches.

**Strengths:**

1. Clear problem framing: LLMs often miss formal action applicability and state updates. The paper targets that gap directly.

2. Explicit verifiability: grounding CoT in VAL feedback forces faithfulness to domain dynamics and mitigates ungrounded story-like CoT.

3. Detailed documentation: The appendix provides extensive implementation details, hyperparameters, prompts, and mathematical formulations, facilitating reproducibility.
4.  Empirical signal: consistent improvements across Blocksworld, Mystery Blocksworld, and Logistics; detailed vs binary feedback ablation is informative.

5. Clear idea & engineering pipeline. Combining structured CoT outputs (explicit ⟨s, a, s′⟩ steps) with an external symbolic verifier (VAL) is a natural and well-motivated approach to close the gap between natural-language reasoning and formal symbolic planning. The proposed two-stage optimization (reasoning loss then final task loss) is intuitively sensible.

**Weaknesses:**

1. Scope: restricted to classical PDDL without conditional effects, temporal/durative actions, or costs; focuses on satisficing rather than optimal planning.
2. Generality: results are on three domains; transfer to richer planning languages and real-world robotics pipelines is untested.
3. Data and compute loop: accuracy improves with more VAL-guided iterations (n), but the cost/benefit curve and stopping criteria are not fully characterized.
4. Comparison set: baselines focus on untuned or instruction-tuned models; a head-to-head against strong LLM-modulo planners or LLMs-as-modelers plus classical solvers (on identical tasks) would sharpen the contribution.
 5. Ablate the effect of modified VAL outputs in training (show effect when you never tamper with VAL).
6. Generalization claims need clearer definition. The paper uses “generalization” loosely; clarify whether this means cross-domain transfer (different domain file), larger problem instances, or semantic obfuscation (Mystery Blocksworld). Provide explicit transfer experiments (train on a set of domains, test on held-out domains).
7.  Test set contamination risk: The paper states "We remove the solution plans from datasets D₂ and D_test" but doesn't clarify if D₂ and D_test contain problems from the same domains with different configurations. If test problems are structurally similar to training problems within the same three domains, this raises concerns about memorization vs. generalization.

**Questions:**

1. You state you sometimes alter VAL outputs to create incorrect explanations for a few plans (Sec. 5.1). Why was this done? How many examples were altered and what safeguards ensure the model does not learn incorrect inference from corrupted feedback?

2. Can you show results for (a) more expressive PDDL features (conditional effects, derived predicates) and/or (b) an additional diverse suite of PlanBench domains? If not possible, explain limitations and expected failure modes.

3. How robust are the CoT traces? Provide examples of unfaithful CoT traces (CoT that looks plausible but the plan fails) and quantify how often your external verifier catches such unfaithful traces. Compare this to literature on CoT unfaithfulness.

4. In result section I see SD value is quite high, why paper have such a high SD value. I assume SD mean standard deviation, correct me If I am wrong. Is it due to high temperature setting of the LLM?

5. How does your approach differ fundamentally from existing work on iterative refinement with external verification (LEPA, STaR, code as symbolic[1], Cot-TL[2])? What is the specific technical contribution beyond applying these ideas to PDDL? Please clarify.

[1]Code-as-Symbolic-Planner: Foundation Model-Based Robot Planning via Symbolic Code Generation

[2] CoT-TL: Low-Resource Temporal Knowledge Representation of Planning Instructions Using Chain-of-Thought Reasoning

---

> ### Author Response · Authors · 2025-11-24
> **Comment on Weaknesses**
>
> We thank the reviewer for the constructive feedback and thorough review. We address the concerns about the weaknesses below:
>
> ## Weakness 1
>
> >  restricted to classical PDDL without conditional effects, temporal/durative actions, or costs; focuses on satisficing rather than optimal planning.
>
> Please refer to common response C3 for a response to this.
>
> ## Weakness 2
>
> > Generality: results are on three domains; transfer to richer planning languages and real-world robotics pipelines is untested.
>
> We acknowledge the limited scope of this work to three domains. However, our primary contribution is demonstrating that LLM planning capabilities can be meaningfully improved through instruction tuning with verification feedback. As our evaluation shows, LLMs currently struggle with PDDL planning even on simple domains, so improving their performance on these well-understood problems is itself a significant contribution. Extending to real-world robotics pipelines for generating task plans is indeed an important next step, but it first requires establishing that LLMs can reliably perform PDDL planning, which this work addresses. We have identified the expansion to additional PlanBench domains as future work (Section 8).
>
> ## Weakness 3
>
> > Data and compute loop: accuracy improves with more VAL-guided iterations (n), but the cost/benefit curve and stopping criteria are not fully characterized.
>
> You raise a valid concern about characterizing the cost-benefit curve of increasing the iteration limit $\eta$. We have acknowledged this as a limitation and future direction (Section 8), noting that dynamic iteration control based on problem complexity would be valuable. Please note that the main focus of this work is establishing that verification-guided instruction tuning improves planning performance; fully characterizing the computational trade-offs is beyond the current scope but remains important for practical deployment.
>
>
> ## Weakness 4
>
> > Comparison set: baselines focus on untuned or instruction-tuned models; a head-to-head against strong LLM-modulo planners or LLMs-as-modelers plus classical solvers (on identical tasks) would sharpen the contribution.
>
> Please refer to common response C1 for a response to this.
>
> ## Weakness 5
>
> > Ablate the effect of modified VAL outputs in training (show effect when you never tamper with VAL).
>
> We appreciate this suggestion, but we respectfully disagree that ablating VAL output tampering is necessary or appropriate. Our modification of VAL outputs was performed only to create negative training examples with specific failure modes that VAL does not natively generate (e.g., incorrect effect application). Deliberately training without correct VAL feedback would undermine the core mechanism of our approach of learning from ground-truth verification signals. We believe this would teach the model incorrect inference patterns rather than provide meaningful ablation insight.
>
> ## Weakness 6
>
> > Generalization claims need clearer definition. The paper uses “generalization” loosely; clarify whether this means cross-domain transfer (different domain file), larger problem instances, or semantic obfuscation (Mystery Blocksworld). Provide explicit transfer experiments (train on a set of domains, test on held-out domains).
>
> We acknowledge that our use of "generalization" was slightly confusing. In the paper we demonstrate multi-domain joint training which is a single model trained on all three domains simultaneously. We renamed the Cross-domain generalization to Multi-domain generalization in Sec. 7. We also add experiments on cross-domain generalization which is training on two domains and testing on a held-out third domain. More details about this are in the common response C4.
>
> ## Weakness 7
>
> >  If test problems are structurally similar to training problems within the same three domains, this raises concerns about memorization vs. generalization.
>
> As clarified in our common response C5. Since test problems are generated with different objects than training problems and were created specifically for this evaluation, memorization is not a concern.

---

> ### Author Response · Authors · 2025-11-24
> **Answer to Questions**
>
> We answer the questions asked by the reviewer below:
>
> ## Question 1
>
> > You state you sometimes alter VAL outputs to create incorrect explanations for a few plans (Sec. 5.1). Why was this done? How many examples were altered and what safeguards ensure the model does not learn incorrect inference from corrupted feedback?
>
> We appreciate the clarification request. When we stated that we "change the VAL's output to add additional effects or remove some effects to have incorrect explanations," we meant the following: we needed to create diverse negative training examples beyond what VAL naturally provides. Specifically, VAL does not natively generate examples where effects are incorrectly added or removed (false positives). To teach the model to detect these failure modes, we augmented VAL's output for correct plans by modifying their explanations to be consistent with incorrect effect patterns. This provided negative examples covering a broader range of failure types. We clarified this methodology in the revision (in Sec. 5.1) to avoid ambiguity about the integrity of our training signal.
>
> ## Question 2
>
> > Can you show results for (a) more expressive PDDL features (conditional effects, derived predicates) and/or (b) an additional diverse suite of PlanBench domains? If not possible, explain limitations and expected failure modes.
>
> Please refer to common response C3
>
>
>
> ## Question 3
>
> > How robust are the CoT traces? Provide examples of unfaithful CoT traces (CoT that looks plausible but the plan fails) and quantify how often your external verifier catches such unfaithful traces. Compare this to literature on CoT unfaithfulness.
>
> Please note that VAL is a formal symbolic verifier. By construction, any CoT trace generated by our model is checked against VAL's ground-truth plan validation. If a CoT trace contains inaccurate reasoning (e.g., incorrect state transitions, unsatisfied preconditions, or failure to reach the goal), VAL will detect and reject it. Therefore, we cannot have "unfaithful" CoT traces in our framework and VAL's verification ensures that every accepted trace is logically consistent with the PDDL semantics. This is a fundamental advantage of our external verification approach compared to methods relying on LLM self-verification. Please refer to VAL [1] for additional details on this.
>
> [1] R. Howey, D. Long and M. Fox, "VAL: automatic plan validation, continuous effects and mixed initiative planning using PDDL," 16th IEEE International Conference on Tools with Artificial Intelligence, Boca Raton, FL, USA, 2004, pp. 294-301, doi: 10.1109/ICTAI.2004.120.
>
> ## Question 4
>
> > In result section I see SD value is quite high, why paper have such a high SD value. I assume SD mean standard deviation, correct me If I am wrong. Is it due to high temperature setting of the LLM?
>
> Our hypothesis is that it happened because we did not maintain a uniform distribution of problem complexities when generating the 100 random test problems for each domain. This variance in problem complexity across the test set contributes to the observed standard deviation. LLMs are known to perform substantially worse on longer-horizon planning problems. When test problems span a range of difficulties, the aggregate accuracy reflects this heterogeneity, resulting in higher SD. A more uniform test set would likely show lower variance but potentially obscure the model's performance across problem complexities. We added the discussion of this effect in the updated paper in Sec. 7.
>
> ## Question 5
>
> > How does your approach differ fundamentally from existing work on iterative refinement with external verification (LEPA, STaR, code as symbolic[1], Cot-TL[2])? What is the specific technical contribution beyond applying these ideas to PDDL? Please clarify.
>
> Please refer to response C2 for comment on this.

---

### Official Review · Reviewer_DDkJ · 2025-11-01

**Soundness:** 3
**Presentation:** 3
**Contribution:** 2
**Rating:** 4
**Confidence:** 4

**Summary:**

This paper introduces PDDL-INSTRUCT, a new method for training LLMs to be much better at symbolic planning in PDDL. The main idea is to teach the model a logical chain-of-thought where it has to explicitly reason about why an action is valid (checking preconditions, applying effects) at every step of a plan. The training has two phases: a basic fine-tuning on PDDL problems and plans, followed by a more advanced phase where the LLM's chain-of-thought plan is checked by an external VAL verifier, and this feedback is used to tune the model even more. PDDL-INSTRUCT shows 94% accuracy on Blocksworld, a 66% jump over the baseline.

**Strengths:**

- The PDDL-INSTRUCT framework is a novel and well-structured approach to instruction tuning that directly targets an LLM's weakness in formal logical verification.
- The paper includes a detailed analysis comparing binary vs. detailed feedback and tests across multiple frontier/public models, which validates the design.

**Weaknesses:**

- It's a good trial, but the results are showing essentially LLMs can still only solve TC0 problems, aligning with the LLM state-tracking literatures. The selected tasks, e.g., BlocksWorld and Logistics, are pretty simple in the planning domain, and some neuro-symbolic methods can perform 100% on them.
- The paper's claim that this could be combined with frameworks like LLM-Modulo to reduce feedback loops is interesting, but where are the results? How is that claim supported?
- The paper is missing many literature references and discussion points from the neuro-symbolic and LLM-Modulo research areas, which are highly relevant.
- The framework's assumptions limit it to simple PDDL features, avoiding common but complex features like conditional effects or durative actions.

**Questions:**

- Could the author provide qualitative examples of failures after tuning? e.g., How does the simple blocksworld planning fail? Is it on long-horizon planning task?

---

> ### Author Response · Authors · 2025-11-24
>
> We thank the reviewer for the feedback and appreciate the thoughtful questions about our approach. We address each concern below.
>
> ## Weakness 1
>
> > It's a good trial, but the results are showing essentially LLMs can still only solve TC0 problems, aligning with the LLM state-tracking literatures. The selected tasks, e.g., BlocksWorld and Logistics, are pretty simple in the planning domain, and some neuro-symbolic methods can perform 100% on them.
>
> Please refer to C1 in the common response and the additional response below
>
> The selection of BlocksWorld and Logistics domains is deliberate as it allows us to establish clear baselines and measure whether LLMs can improve on these well-understood problems. Our results show substantial improvements (66% absolute gain over baseline on BlocksWorld), demonstrating that PDDL-Instruct is an effective approach to enhancing LLM planning capabilities. We acknowledge that LLMs still fall short of symbolic methods, and we do not claim otherwise. However, the contribution is in understanding how to improve LLM planning through instruction-driven learning with verification feedback. We will clarify this positioning in the revision to emphasize that we are studying LLM improvement, not LLM replacement of symbolic approaches.
>
>
> ## Weakness 2
>
> > The paper's claim that this could be combined with frameworks like LLM-Modulo to reduce feedback loops is interesting, but where are the results? How is that claim supported?
>
> You raise a valid concern about the empirical support for our LLM-Modulo integration claim. We added experiments in Sec. D.4 to support our claim. As mentioned in the common response  PDDLInstruct with LLM Modulo framework makes the overall LLM Modulo solution upto 3.6 - 7.1 times faster (Appendix D.4) on larger models. This can also be explained theoretically in addition to the empirical results. As described in the LLM-Modulo paper, the framework operates as a Generate-Test-Critique loop where the LLM generates candidate plans, critics (such as VAL) test them, and feedback guides refinement. A key observation is that if the LLM generates more accurate candidates, the Test-Critique phases will identify fewer errors, thereby reducing the number of iterations required before finding a valid plan. Since PDDL-Instruct directly improves the quality of LLM-generated plans (achieving 94% accuracy on BlocksWorld vs. 28% baseline), an LLM tuned by our approach would produce fewer erroneous plans and thus require fewer Generate-Test-Critique iterations when deployed in the LLM-Modulo framework. This reduction is a direct theoretical consequence of improved LLM accuracy.
>
> ## Weakness 3
>
> > The paper is missing many literature references and discussion points from the neuro-symbolic and LLM-Modulo research areas, which are highly relevant.
>
> We added the suggested reference in Sec. 2. We would appreciate it if the reviewer could point us to specific neuro-symbolic approaches and LLM-Modulo variants that they feel should also be discussed, so that we can incorporate them appropriately in the related work section of the revision.
>
> Please refer to C2 in the common response
>
> ## Weakness 4
>
> > The framework's assumptions limit it to simple PDDL features, avoiding common but complex features like conditional effects or durative actions.
>
> Please refer to C3 in the common response
>
> ## Question 1
>
> > Could the author provide qualitative examples of failures after tuning? e.g., How does the simple blocksworld planning fail? Is it on long-horizon planning task?
>
> Thank you for this question. This was already captured in Sec. D.2 in the original submission. We added a discussion for it now, and also added a new research question RQ4 to highlight this, and added Table 2, along with a discussion for it to address this.

---

### Official Review · Reviewer_eDNw · 2025-11-03

**Soundness:** 3
**Presentation:** 3
**Contribution:** 3
**Rating:** 6
**Confidence:** 4

**Summary:**

The authors propose an instruction-following tuning framework: PDDL-Instruct to improve the symbolic planning capabilities of LLMs using chain-thought reasoning. The solution decomposes planning verification into atomic reasoning steps incorporating the resulting structure for instruction-tuning. PDDL-Instruct  integrates external verification feedback from a plan validator (i.e. VAL by Howey et al. (2004)) during training to guide and refine the model's planning and reasoning outputs. Experimental results on three different planning domains from PlanBench show that the tuning paradigm introduced by PDDL-Instruct results in more capable planning models, substantially outperforming naive instruction tuning baselines.

**Strengths:**

- The proposed framework results in significant improvements in the underlying LLMs' planning capabilities across different symbolic planning domains.
- The use of an external plan validator, VAL, is well-aligned with the challenges of the domain, avoiding over-reliance on LLMs' imperfect self-reflection.
- The paper responsibly acknowledges the limitations of the proposed framework, such as restricted PDDL feature coverage, and outlines promising avenues for improving self-verification capabilities and broader domain coverage.

**Weaknesses:**

- The paper primarily compares the proposed PDDL-Instruct framework against naive instruction tuning baselines, where the LLMs are fine-tuned without the logical CoT and verification feedback mechanisms. It would be useful to compare or at least relate the performance of PDDL-Instruct to other symbolic planners tested on the selected domains from PlanBench.
- While splitting training into initial and CoT instruction phases boosts the end performance, the paper would benefit from clearer explanation and explicit details regarding test set construction.
    - How are test problems selected and controlled for overlap or similarity with training domains and instances to ensure cross-domain generalisation evaluation?
- The paper would benefit from a more detailed analysis of employing the specialised loss functions used in the experiments against more conventional alternatives in this space (e.g., against the negative loglikelihood instead of the $\mathcal{L}_{\text{plan}}$, as described in Appendix B.2.2). Such comparison could provide insight into the impact and necessity of the designed loss components.

**Questions:**

- As far as I understand, a separate model was trained for each benchmark, including all relevant training phases. It would be interesting to explore how a model jointly trained across all three domains would perform.

---

> ### Author Response · Authors · 2025-11-24
>
> We thank the reviewer for the feedback and appreciate the constructive suggestions. We address each concern below.
>
> ## Weakness 1
>
>
> > It would be useful to compare or at least relate the performance of PDDL-Instruct to other symbolic planners tested on the selected domains from PlanBench.
>
> Please refer to C1 in common response
>
> ## Weakness 2
>
> >  the paper would benefit from clearer explanation and explicit details regarding test set construction. How are test problems selected and controlled for overlap or similarity with training domains and instances to ensure cross-domain generalisation evaluation?
>
> Please refer to common response C5
>
>
> ## Weakness 3
>
>
> > The paper would benefit from a more detailed analysis of employing the specialised loss functions used in the experiments against more conventional alternatives in this space (e.g., against the negative loglikelihood instead of the , as described in Appendix B.2.2)
>
> We appreciate the question about our specialized loss functions compared to conventional alternatives. We would like to make an important clarification that the Binary Cross-Entropy (BCE) loss used in $L_{final} $ is mathematically equivalent to negative log-likelihood, and we employ it specifically to make the discrete loss signals from other components of the loss function (such as the state distance and feedback penalties in $L_{reasoning}$) differentiable and trainable. The specialized loss functions we designed encode domain knowledge that planning experts have developed over decades: checking precondition satisfaction, tracking state effects, and detecting goal achievement. By explicitly incorporating these concepts into our loss function, we guide the model learning toward planning-relevant objectives rather than relying on generic language modeling losses.
>
> ## Question 1
>
> > As far as I understand, a separate model was trained for each benchmark, including all relevant training phases. It would be interesting to explore how a model jointly trained across all three domains would perform.
>
> Please refer to common response C4.

---

### Author Response · Authors · 2025-11-24
**Paper Updates**

We thank the reviewers for their constructive feedback. They highlighted several key strengths of our work, including the significant improvements in planning capabilities across domains, the use of external verification to avoid over-reliance on LLM self-reflection, and responsible acknowledgment of limitations.

Based on the reviews, we made the following changes to the paper:

1. Added accuracy analysis as a function of problem complexity. We framed this analysis as a response to “RQ4: How robust are the models to problem complexity? Does performance degrade as it increases?”. The analysis is added at the end of Sec. 7, with results summarized in Table 2.

2. Added preliminary cross-domain transfer experiments (training on 2 domains, testing on held-out 3rd): (Sec D.3 in the supplement). We also renamed the solution to RQ3 as multi-domain generalization instead of cross-domain generalization to account for feedback on misclarification by Reviewer arPJ.

3. Explicitly contrasted our parameter-update approach with prompting-based methods like LEPA and Code-as-Symbolic-Planner (Sec 2, after LLM Modulo subsection)

4. Added Fast Downward’s 100% accuracy to Table 1 for proper contextualization

5. Included explanation of high standard deviation in results (problem complexity variance), and analysis of results presented in Sec. D.1 and D.2.

6. Emphasized single multi-domain model training in Sec. 7. Also highlighted it under main contributions in Sec. 1.

7. Added emphasis on parameter-update paradigm as core novelty under main contributions in Sec. 1.

8. Added experiments in Sec. D.4 to show that integration of PDDLInstruct with LLM Modulo framework leads to faster solutions by LLM Modulo as compared to using an untuned LLM with it.

9. We also added a diff file in the supplementary to highlight the exact changes we made in the paper.

---

> ### Author Response · Authors · 2025-11-24
> **Comments on Weaknesses and Questions**
>
> ## C1. About comparison with Symbolic Planners
>
> The goal of this work is not to replace symbolic planners like Fast Downward (FD) [1], which necessarily outperform LLMs on well-defined planning problems and achieve 100% accuracy in our experiments. Rather, we aim to demonstrate that LLM planning capabilities can be meaningfully improved through systematic instruction tuning combined with verification feedback. This is important to address the prior perception that LLMs cannot do symbolic planning [2,3,4,5]. Given that planning is PSPACE-complete, symbolic planners like FD may struggle to scale to complex real-world domains. In the revision, we have added FD's 100% accuracy to Table 1 to properly contextualize our improvements. Additionally, improving LLM planning enables the same model (LLM) to handle planning, natural language, code generation, and other tasks in a unified framework.
>
> ## C2. About comparison with LLM Modulo Approaches
>
> PDDL-Instruct differs fundamentally from neuro-symbolic approaches: we improve LLMs through parameter updates during training, not through hybrid architectures combining neural networks with symbolic engines. This parameter-update paradigm allows models to learn from verification feedback across iterations, which is a novel contribution to LLM based planning. Critically, our approach preserves the LLM's generality. After instruction tuning, the model remains capable of text generation, code writing, and other tasks. In contrast, specialized frameworks like LLM-Modulo build task-specific hybrid architectures that lose this flexibility. When combined with LLM-Modulo, PDDL-Instruct reduces required iterations by 3.6 - 7.1 times (Appendix D.4) on larger models. We restructured Sec. 2 with updated references to clarify this distinction, and added Appendix D.4 with the LLM-Modulo experiments.
>
> ## C3. About limitation of PDDL features covered
>
> We acknowledge this limitation and have already identified it in the paper (lines 466-469 in orginal paper) in our limitations and future work discussion. As mentioned in lines 224-226, the simplification to avoid complex PDDL features such as conditional effects and durative actions was made deliberately to keep reasoning chains tractable for LLMs. We recognize that this limits practical applicability to real-world planning problems
>
> ## C4. Cross domain generalization
>
> We appreciate this question coming from the reviewers and realize our experimental setup was not clearly communicated. To clarify, we train a single model per architecture (one Llama-3 model, one GPT-4 model, one Gemma-3 model) on training examples from all three domains simultaneously. We do not train separate models for each domain. This demonstrates that our instruction tuning approach can handle multi-domain generalization, the model learns to apply planning reasoning across structurally different domains without domain-specific finetuning. Moreover, our new cross-domain generalization experiments in Sec. D.3 (suggested by reviewer gPN4) reveals an encouraging finding that when we removed Mystery Blocksworld from the training set entirely and trained only on Blocksworld and Logistics, we observed minimal performance degradation on Mystery Blocksworld. This modest 3-4 percentage point gap demonstrates the model learns generalizable reasoning rather than memorizing domain-specific patterns. We make this multi-domain learning contribution explicit in the revision and highlight this generalization result as evidence that our instruction tuning approach develops domain-agnostic planning capabilities.
>
> ## C5. Training and Test sets construction
>
> We appreciate this question and recognize that our description of dataset construction was unclear. PlanBench provides extensive generators ensuring test problems are genuinely new. Domain files are identical across train/test splits (same domain rules), while problem files contain unique object instantiations. This prevents data leakage while maintaining domain structure.  We added this additional information in Sec. 6. For a detailed explanation of PlanBench's generation methodology, we refer readers to PlanBench paper [5].
>
> ---------
> ## References:
>
> [1] Helmert M. The fast downward planning system. Journal of Artificial Intelligence Research. 2006.
>
> [2] Valmeekam K, Marquez M, Sreedharan S, Kambhampati S. On the planning abilities of large language models-a critical investigation. NeurIPS 2023
>
> [3] Stechly K, Valmeekam K, Kambhampati S. On the self-verification limitations of large language models on reasoning and planning tasks. ICLR 2025
>
> [4] Huang S, Cohn T, Lipovetzky N. Chasing progress, not perfection: Revisiting strategies for end-to-end LLM plan generation. ICAPS 2025
>
> [5] Valmeekam K, Marquez M, Olmo A, Sreedharan S, Kambhampati S. Planbench: An extensible benchmark for evaluating large language models on planning and reasoning about change. NeurIPS 2023

---

### Meta-Review · Area_Chair_jS3y · 2026-01-07

**Summary:**

The paper introduces PDDL-Instruct, an instruction tuning framework that improves LLM planning capabilities through chain-of-thought reasoning and external verification feedback from VAL. Reviewers recognized several strengths: significant empirical improvements, principled use of external verification rather than unreliable LLM self-reflection, and transparent acknowledgment of limitations. However, all four reviewers raised concerns about the evaluation scope and positioning. The primary issues included: lack of comparison with symbolic planners and LLM-Modulo frameworks, restriction to simple PDDL features without conditional effects or durative actions, evaluation on only three domains with unclear generalization properties, and high standard deviations in reported results requiring explanation. Reviewers also questioned whether the contribution was primarily integrative rather than introducing fundamentally new techniques.

**Reviewer Concerns:**

The authors addressed several key concerns effectively. The new cross-domain transfer experiments (Sec D.3) showing minimal degradation when training on two domains and testing on a held-out third directly addresses generalization questions from Reviewers gPN4 and arPJ. The LLM-Modulo integration experiments (Sec D.4) demonstrating 3.6-7.1x speedup provide empirical support for previously unsupported claims, addressing Reviewer DDkJ's concern. The clarification that a single model is trained across all domains simultaneously, not separate per-domain models, resolves confusion about multi-domain learning.

Outstanding concerns remain regarding the restricted PDDL feature coverage. The authors acknowledge this limitation but provide no experimental evidence that the approach extends to conditional effects, derived predicates, or temporal planning. The positioning relative to prompting-based methods like LEPA and Code-as-Symbolic-Planner remains somewhat superficial despite added discussion. The high standard deviation explanation (problem complexity variance) is reasonable but points to a methodological weakness in test set construction that could have been addressed with stratified sampling.

**Reviewer Scores:**

Reviewers would likely retain their scores.

---

### Decision · Program_Chairs · 2026-01-26

Reject